# Constrained Bi-Level Optimization: Proximal Lagrangian Value function Approach and Hessian-free Algorithm

**Wei Yao**[124], **Chengming YU**[1], **Shangzhi Zeng**[31], **Jin Zhang**[21*]

[1]National Center for Applied Mathematics Shenzhen, SUSTech, [2]Department of Mathematics, SUSTech, [3]Department of Mathematics and Statistics, UVic, [4]CETC Key Laboratory of Smart City Modeling Simulation and Intelligent Technology, The Smart City Research Institute of CETC

`{yaow,zhangj9}@sustech.edu.cn`, `yucm@mail.sustech.edu.cn`,
`zengshangzhi@uvic.ca`

## Abstract

This paper presents a new approach and algorithm for solving a class of constrained Bi-Level Optimization (BLO) problems in which the lower-level problem involves constraints coupling both upper-level and lower-level variables. Such problems have recently gained significant attention due to their broad applicability in machine learning. However, conventional gradient-based methods unavoidably rely on computationally intensive calculations related to the Hessian matrix. To address this challenge, we devise a smooth proximal Lagrangian value function to handle the constrained lower-level problem. Utilizing this construct, we introduce a single-level reformulation for constrained BLOs that transforms the original BLO problem into an equivalent optimization problem with smooth constraints. Enabled by this reformulation, we develop a Hessian-free gradient-based algorithm—termed proximal Lagrangian Value function-based Hessian-free Bilevel Algorithm (LV-HBA)—that is straightforward to implement in a single loop manner. Consequently, LV-HBA is especially well-suited for machine learning applications. Furthermore, we offer non-asymptotic convergence analysis for LV-HBA, eliminating the need for traditional strong convexity assumptions for the lower-level problem while also being capable of accommodating non-singleton scenarios. Empirical results substantiate the algorithm's superior practical performance.

## 1 Introduction

In this work, we consider the constrained Bi-Level Optimization (BLO) problems with possibly coupled lower-level (LL) constraints, which is in form of,

$$\min_{x \in X, y \in Y} F(x,y) \quad \text{s.t.} \quad y \in S(x), \tag{1}$$

where $S(x)$ denotes the set of optimal solutions for the constrained LL problem,

$$\min_{y \in Y} f(x,y) \quad \text{s.t.} \quad g(x,y) \leq 0. \tag{2}$$

Here both $X \subseteq \mathbb{R}^n$ and $Y \subseteq \mathbb{R}^m$ are closed convex sets. The upper-level (UL) objective $F : X \times Y \to \mathbb{R}$, the LL objective $f : X \times Y \to \mathbb{R}$, and the LL constraint mapping $g : X \times Y \to \mathbb{R}^l$ are continuously differentiable functions. It is noteworthy that both LL objective $f$ and constraint mapping $g$ are functions of UL variable $x$ LL variable $y$.

BLO has recently emerged as a powerful tool for tackling various modern machine learning problems characterized by inherent hierarchical structures, such as hyperparameter optimization Pedregosa (2016); Franceschi et al. (2018); Mackay et al. (2019), meta learning Franceschi et al. (2018); Zügner & Günnemann (2019); Rajeswaran et al. (2019); Ji et al. (2020), neural architecture search Liu et al. (2018); Liang et al. (2019); Elsken et al. (2020), to name a few. Among them, the constrained BLOs capture several important applications, including adversarial learning

---

*Correspondence to Jin Zhang (zhangj9@sustech.edu.cn)

Table 1: Comparison of our method LV-HBA with closely related works for **constrained BLO** ( IG-AL Tsaknakis et al. (2022), SIGD Khanduri et al. (2023), AiPOD/E-AiPOD Xiao et al. (2023b), GAM Xu & Zhu (2023), BVFSM Liu et al. (2023a) ). All method, except BVFSM, require the LL strongly convexity assumption, which implies the singleton of the LL minimizer. GAM and BVFSM provide only asymptotic convergence analysis. Using $h(x,y) = 0 \Leftrightarrow h(x,y) \le 0, -h(x,y) \le 0$, our method allows for the inclusion of linear constraints.

| Method | LL Objective | LL Constraints | Hessian-Free | Single Loop | Non-Singleton |
|--------|--------------|----------------|--------------|-------------|---------------|
| IG-AL | Strongly Convex | $Ay \le b$ | ✗ | ✗ | ✗ |
| SIGD | Strongly Convex | $Ay \le b$ | ✗ | ✗ | ✗ |
| AiPOD E-AiPOD | Strongly Convex | $Ay + h(x) = c$ | ✗ | ✔ | ✗ |
| GAM | Strongly Convex | $g(x,y) \le 0$ $h(x,y) = 0$ | ✗ | ✗ | ✗ |
| BVFSM | Convex | $g(x,y) \le 0$ | ✔ | ✗ | ✔ |
| LV-HBA | Convex | $g(x,y) \le 0$ | ✔ | ✔ | ✔ |

Madry et al. (2018); Wong et al. (2019); Zhang et al. (2022), federated learning Fallah et al. (2020); Tarzanagh et al. (2022); Yang et al. (2023b), see the recent survey papers Liu et al. (2021a); Zhang et al. (2023) for more applications in machine learning and signal processing.

Owing to their effectiveness and scalability, gradient-based algorithms have become mainstream techniques for BLO in learning and vision fields Liu et al. (2021a). While gradient-based algorithms for unconstrained BLO problems have been extensively explored in the literature Ghadimi & Wang (2018); Shaban et al. (2019); Liu et al. (2020; 2021b); Huang et al. (2022); Ji et al. (2021; 2022); Hong et al. (2023); Dagréou et al. (2022); Ye et al. (2022); Liu et al. (2023b); Kwon et al. (2023a), research focusing on efficient methods for constrained BLO problems is quite limited. This gap is especially evident in scenarios where LL constraints couple both UL and LL variables.

Indeed, the majority of existing works in this direction focus on particular types of constrained LL problems. For instance, recent works Tsaknakis et al. (2022); Khanduri et al. (2023) address constrained BLOs where LL problem pertains to minimizing a strongly convex objective subject to linear inequality constraints. The study Xiao et al. (2023a) considers the stochastic BLO problems with equality constraints at both upper and lower levels, while Xu & Zhu (2023) studies BLOs wherein LL problem is convex with equality and inequality constraints and presumes that LL objective is strongly convex and the constraints satisfy Linear Independence Constraint Qualification (LICQ). Notably, the methods presented in these works all employ implicit gradient-based techniques, relying on implicit gradient computation of LL solution mapping. This dependency requires both the uniqueness and smoothness of LL solution mapping, thereby limiting its applicability. Critically, implicit gradient techniques necessitate computationally intensive calculations related to LL Hessian matrix. In this context, a natural yet important question is: *Can we devise Hessian-free algorithms for constrained BLOs?*

A recent affirmation to this question is provided in Liu et al. (2023a), leveraging the value function approach Ye & Zhu (1995). For constrained BLOs, value function-based methods encounter issues tied to non-differentiable constraints emerging from the value function-based reformulation. To circumvent this non-smoothness issue, Liu et al. (2023a) introduces a sequential approximation minimization strategy. Herein, quadratic regularization alongside penalty/barrier functions of LL inequality constraints are applied to smooth LL value function. Nonetheless, this approach necessitates solving a series of subproblems and lacks a non-asymptotic analysis. This leads to a practical question: *Can we devise a Hessian-free algorithm in a single-loop manner for constrained BLOs?*

## 1.1 MAIN CONTRIBUTIONS

In this study, we provide an affirmative answer to the previously raised question. We first propose a single-level reformulation for constrained BLO problems by defining a proximal Lagrangian value function associated with the constrained LL problem. This function is defined as the value

function of a strongly-convex-strongly-concave proximal min-max problem and exhibits continuous differentiability. As a result, our approach recasts constrained BLO problems into single-level optimization problems with smooth constraints. Drawing from this reformulation, we devise a Hessian-free gradient-based algorithm for constrained BLOs, and provide the non-asymptotic convergence analysis. Conducting such an analysis, especially given LL constraints, is non-trivial. By utilizing the strongly-convex-strongly-concave structure within the min-max problem of the proximal Lagrangian value function, we can approximate its gradient using only the first-order data from LL problem. The error in this gradient approximation remains controllable, without the need for LL objective's strong convexity. This facilitates our establishment of the non-asymptotic convergence analysis of the proposed algorithm for constrained LL problem with a merely convex LL objective.

Our primary contributions are outlined below.

- We introduce a novel proximal Lagrangian value function to handle constrained LL problem. By leveraging this function, we present a new single-level reformulation for constrained BLOs, converting them into equivalent single-level optimization problems with smooth constraints.

- Drawing from our reformulation, we propose the proximal Lagrangian Value function-based Hessian-free Bi-level Algorithm (LV-HBA) tailored for constrained BLO problems with LL constraints coupling both UL and LL variables. To our knowledge, this work is the first to develop a provably Hessian-free gradient-based algorithm for constrained BLOs in a single-loop manner. A brief summary of the comparison of LV-HBA with closely related works is provided in Table 1.

- We rigorously establish the non-asymptotic convergence analysis of LV-HBA. Employing the proximal Lagrangian value function, we eliminate the necessity for LL objective's strong convexity, thereby accommodating merely convex LL scenarios.

- We evaluate the efficiency of LV-HBA through numerical experiments on synthetic problems, hyperparameter optimization for SVM and federated bilevel learning. Empirical results validate the superior practical performance of LV-HBA.

## 1.2 RELATED WORK

In the section we give a brief review of some recent works that are directly related to ours. An expanded review of recent studies on BLOs is provided in Section A.2.

**Reformulations for BLOs.** One of the most commonly approaches for BLOs is to reformulate them as single-level problems. This can often be done in two ways Dempe & Zemkoho (2013). One is known as KKT reformulation Kim et al. (2020), which replaces LL problem with its Karush-Kuhn-Tucker (KKT) conditions. Consequently, it unavoidably relies on first-order gradient information. As a result, gradient-based algorithms based on it also necessitate second-order gradient information. In contrast, value function reformulation does not rely on any gradient information. Benefiting from this, the majority of existing Hessian-free gradient-based algorithms for both unconstrained and constrained BLOs, are developed based on value function reformulation, see, e.g., Liu et al. (2021b; 2023a); Ye et al. (2022); Sow et al. (2022); Shen & Chen (2023); Kwon et al. (2023a); Lu & Mei (2023); Lu (2024). Recently, Gao et al. (2023) proposes a new reformulation of BLOs, using Moreau envelope of LL problem, to weaken the underlying assumption from LL full convexity in Gao et al. (2022) to weak convexity. However, Moreau envelope-based reformulation still encounters challenges related to non-differentiable constraints, arising from the reformulation itself.

**Algorithms for Constrained BLOs.** Other than the previously mentioned works that focus on LL constraints, there is a line of research dedicated to addressing the constrained UL setting, including: implicit approximation methods in Ghadimi & Wang (2018); two-timescale framework in Hong et al. (2023); single-timescale method in Chen et al. (2022a); initialization auxiliary method in Liu et al. (2021c); Bregman distance-based method in Huang et al. (2022); proximal gradient-type algorithm in Chen et al. (2022b); inexact conditional gradient method in Abolfazli et al. (2023).

## 2 PROXIMAL LAGRANGIAN VALUE FUNCTION APPROACH

In this section, we introduce a novel single-level reformulation for constrained BLO problems, foundational to our proposed methodology. Furthermore, we describe the proposed algorithm LV-HBA. To simplify our notation, throughout this paper, for LL constraint mapping $g : X \times Y \to \mathbb{R}^l$ and any vectors $\lambda \in \mathbb{R}^l_+, z \in \mathbb{R}^l$ , we represent $\sum_{i=1}^l \lambda_i g_i$ and $\sum_{i=1}^l z_i g_i$ as $\lambda g$ and $zg$, respectively. Similarly, $\sum_{i=1}^l \lambda_i \nabla_y g_i$ and $\sum_{i=1}^l z_i \nabla_y g_i$ are denoted by $\lambda \nabla_y g$ and $z \nabla_y g$, respectively.

### 2.1 REFORMULATION VIA PROXIMAL LAGRANGIAN VALUE FUNCTION

We start by introducing the proximal Lagrangian value function $v_\gamma(x, y, z)$ for LL problem, drawing inspiration from Moreau envelope value function discussed in Gao et al. (2023). It is defined as:

$$v_\gamma(x, y, z) := \min_{\theta \in Y} \max_{\lambda \in \mathbb{R}^l_+} \left\{ f(x, \theta) + \lambda g(x, \theta) + \frac{1}{2\gamma_1} \|\theta - y\|^2 - \frac{1}{2\gamma_2} \|\lambda - z\|^2 \right\}, \tag{3}$$

where $z \in \mathbb{R}^l$, $\mathbb{R}^l_+ := \{\lambda \in \mathbb{R}^l | \lambda_i \geq 0 \ \forall i\}$, $\gamma := (\gamma_1, \gamma_2) \geq 0$ is the proximal parameter. Note that $f(x, y) + \lambda g(x, y)$ is the Lagrangian function of LL problem. Employing this function, we present a smooth reformulation for constrained BLO problem (1):

$$\min_{(x,y) \in C, z \geq 0} F(x, y) \quad \text{s.t.} \quad f(x, y) - v_\gamma(x, y, z) \leq 0, \tag{4}$$

where $C := \{(x, y) \in X \times Y \mid g(x, y) \leq 0\}$. Under the convexity of LL problem and the existence of multipliers for LL problem, the equivalence between reformulation (4) and constrained BLO problem (1) is established. Notably, $f(x, y) - v_\gamma(x, y, z) \geq 0$ for any $(x, y, z) \in C \times \mathbb{R}^l_+$. Comprehensive proofs can be found in Theorem A.1 in Appendix A.3.

To guarantee the theoretical convergence of the proposed method, instead of directly solving reformulation (4), we consider its variant using a truncated proximal Lagrangian value function,

$$v_{\gamma,r}(x, y, z) := \min_{\theta \in Y} \max_{\lambda \in Z} \left\{ f(x, \theta) + \lambda g(x, \theta) + \frac{1}{2\gamma_1} \|\theta - y\|^2 - \frac{1}{2\gamma_2} \|\lambda - z\|^2 \right\}, \tag{5}$$

where $Z := [0, r]^l \subseteq \mathbb{R}^l_+$ and $r > 0$. Compared with $v_\gamma(x, y, z)$, the truncated version $v_{\gamma,r}(x, y, z)$ is defined by maximizing $\lambda$ over a bounded set $Z$ instead of over $\mathbb{R}^l_+$. And the truncated proximal Lagrangian value function gives us the following variant to reformulation (4),

$$\min_{(x,y) \in C, z \in Z} F(x, y) \quad \text{s.t.} \quad f(x, y) - v_{\gamma,r}(x, y, z) \leq 0. \tag{6}$$

Note that $f(x, y) - v_{\gamma,r}(x, y, z) \geq 0$ for any $(x, y, z) \in C \times Z$. If $r$ is sufficiently large, the solution of reformulation (4) can be obtained by solving variant (6). A comprehensive proof is presented in Theorem A.2 within Appendix A.3.

### 2.2 GRADIENT OF PROXIMAL LAGRANGIAN VALUE FUNCTION

An important property of $v_\gamma(x, y, z)$ and its truncated counterpart $v_{\gamma,r}(x, y, z)$ is their continuous differentiability under the setting of this study, as elucidated in Assumptions 3.2 and 3.3. Specifically, if $f(x, \cdot)$ and $g(x, \cdot)$ are convex on $Y$, the proximal min-max problems in (3) and (5) are both strongly-convex-strongly-concave. By invoking saddle point theorem, these problems possess unique saddle points. Moreover, with continuous differentiability for both $f$ and $g$, the gradient $\nabla v_{\gamma,r}(x, y, z)$ can be derived with the explicit expression as

$$\nabla v_{\gamma,r}(x, y, z) = \left( \nabla_x f(x, \theta^*) + \lambda^* \nabla_x g(x, \theta^*), \frac{(y - \theta^*)}{\gamma_1}, \frac{(\lambda^* - z)}{\gamma_2} \right), \tag{7}$$

where $(\theta^*, \lambda^*) := (\theta^*(x, y, z), \lambda^*(x, y, z))$ denotes the unique saddle point for the min-max problem in (5). Similarly, for $v_\gamma(x, y, z)$, the gradient $\nabla v_\gamma(x, y, z)$ shares the same form as in (7), but with $(\theta^*, \lambda^*)$ corresponding to the unique saddle point of the min-max problem in (3). A detailed proof can be found in Lemma A.1 within the Appendix.

### 2.3 THE PROPOSED ALGORITHM

We introduce LV-HBA, a Hessian-free gradient-based algorithm designed for constrained BLOs.

At each iteration, given the current values of $(x^k, y^k, z^k, \theta^k, \lambda^k)$, we initiate by executing a single gradient descent ascent (GDA) step for the proximal min-max problem described in (5), updating the variables $(\theta, \lambda)$ as follows

$$(\theta^{k+1}, \lambda^{k+1}) = \mathrm{Proj}_{Y \times Z}\left((\theta^k, \lambda^k) - \eta_k(d_\theta^k, d_\lambda^k)\right), \tag{8}$$

where $\mathrm{Proj}_Z$ represents the Euclidean projection onto the bounded box $Z$, and

$$(d_\theta^k, d_\lambda^k) := \left(\nabla_y f(x^k, \theta^k) + \lambda^k \nabla_y g(x^k, \theta^k) + \frac{1}{\gamma_1}(\theta^k - y^k), -g(x^k, \theta^k) + \frac{1}{\gamma_2}(\lambda^k - z^k)\right). \tag{9}$$

Subsequently, we update the variables $(x, y, z)$ as follows

$$\begin{aligned}(x^{k+1}, y^{k+1}) &= \mathrm{Proj}_C\left((x^k, y^k) - \alpha_k(d_x^k, d_y^k)\right), \\ z^{k+1} &= \mathrm{Proj}_Z\left(z^k - \beta_k d_z^k\right),\end{aligned} \tag{10}$$

where the directions are defined as:

$$\begin{aligned} d_x^k &:= \frac{1}{c_k}\nabla_x F(x^k, y^k) + \nabla_x f(x^k, y^k) - \nabla_x f(x^k, \theta^{k+1}) - \lambda^{k+1}\nabla_x g(x^k, \theta^{k+1}), \\ d_y^k &:= \frac{1}{c_k}\nabla_y F(x^k, y^k) + \nabla_y f(x^k, y^k) - \frac{1}{\gamma_1}(y^k - \theta^{k+1}), \\ d_z^k &:= -\frac{1}{\gamma_2}(\lambda^{k+1} - z^k). \end{aligned} \tag{11}$$

A comprehensive description of LV-HBA is provided in Algorithm 1.

---

**Algorithm 1** proximal Lagrangian Value function-based Hessian-free Bi-level Algorithm (LV-HBA)

---

**Initialize:** $(x^0, y^0) \in X \times Y$, $z^0 \in Z$, $(\theta^0, \lambda^0) \in Y \times \mathbb{R}_+^l$, stepsizes $\alpha_k, \beta_k, \eta_k$, proximal parameter $\gamma$, penalty parameter $c_k$;

1: **for** $k = 0, 1, \ldots, K - 1$ **do**
2:     calculate $(d_\theta^k, d_\lambda^k)$ as in equation (9);
3:     update $(\theta^{k+1}, \lambda^{k+1}) = \mathrm{Proj}_{Y \times Z}\left((\theta^k, \lambda^k) - \eta_k(d_\theta^k, d_\lambda^k)\right)$;
4:     calculate $d_x^k, d_y^k, d_z^k$ as in equation (11);
5:     update

$$\begin{aligned}(x^{k+1}, y^{k+1}) &= \mathrm{Proj}_C\left((x^k, y^k) - \alpha_k(d_x^k, d_y^k)\right), \\ z^{k+1} &= \mathrm{Proj}_Z\left(z^k - \beta_k d_z^k\right).\end{aligned}$$

6: **end for**

---

We provide insight into the construction of LV-HBA. The update of variables $(x, y, z)$ in (10) can be interpreted as an inexact alternating proximal gradient step from $(x^k, y^k, z^k)$ concerning the following minimization problem:

$$\min_{(x,y) \in C, z \in Z} \frac{1}{c_k}F(x, y) + f(x, y) - v_{\gamma, r}(x, y, z).$$

Drawing from the gradient formula of $v_{\gamma, r}(x, y, z)$ given in (7), $(\nabla_x f(x^k, \theta^{k+1}) + \lambda^{k+1}\nabla_x g(x^k, \theta^{k+1}), (y^k - \theta^{k+1})/\gamma_1, (\lambda^{k+1} - z^k)/\gamma_2)$ in (11) can be considered as approximations to $\nabla v_{\gamma, r}(x, y, z)$, using $(\theta^{k+1}, \lambda^{k+1})$ as a proxy to $(\theta^*, \lambda^*)$.

Particularly, when the feasible sets $Y$ and $C$ exhibit "projection-friendly" characteristics, meaning that the Euclidean projection onto them is computationally efficient, LV-HBA can be characterized as a single-loop Hessian-free gradient-based algorithm for constrained BLO problems.

## 3 NON-ASYMPTOTIC CONVERGENCE ANALYSIS

In this section, we conduct a non-asymptotic analysis for LV-HBA. We begin by outlining the basic assumptions adopted throughout this work.

### 3.1 GENERAL ASSUMPTIONS

The following assumptions formalize the smoothness property of UL objective $F$, and smoothness and convexity properties of LL objective $f$ and LL constraints $g$.

**Assumption 3.1** (**Upper-Level Objective**). *The UL objective $F$ is $L_F$-smooth* on $X \times Y$. Additionally, $F$ is bounded below on $X \times Y$, i.e., $\underline{F} := \inf_{(x,y) \in X \times Y} F(x,y) > -\infty$.*

**Assumption 3.2** (**Lower-Level Objective**). *Assume that the following conditions hold:*

(i) *$f$ is convex w.r.t. LL variable $y$ on $Y$ for any $x \in X$.*

(ii) *$f$ is continuously differentiable on an open set containing $X \times Y$ and is $L_f$-smooth on $X \times Y$.*

Given that $f$ is $L_f$-smooth on $X \times Y$, by leveraging the descent lemma (Beck, 2017, Lemma 5.7), it can be deduced that $f$ is also $L_f$-weakly convex, i.e, $f(x,y) + L_f \|(x,y)\|^2/2$ is convex on $X \times Y$. Consequently, under Assumption 3.2, $f$ is $\rho_f$-weakly convex on $X \times Y$, with $\rho_f \geq 0$ potentially being smaller than $L_f$. To precisely determine the range for the step sizes of LV-HBA , we will employ the weak convexity constant of $f$, $\rho_f$, in subsequent results.

**Assumption 3.3** (**Lower-Level Constraints**). *Assume that the following conditions hold:*

(i) *$g(x,y)$ is convex and be $L_g$-Lipschitz continuous on $X \times Y$.*

(ii) *$g(x,y)$ is continuously differentiable on an open set containing $X \times Y$, $\nabla_x g(x,y)$ and $\nabla_y g(x,y)$ are $L_{g_1}$ and $L_{g_2}$-Lipschitz continuous on $X \times Y$, respectively.*

Our assumptions are based solely on the first-order differentiability of the problem data. The setting of this study substantially relaxes the existing requirement for second-order differentiability in constrained BLO literature. Notably, we do not impose strong convexity on the LL objective $f$. As a result, our analysis encompasses LL problem non-singleton scenarios, as detailed in Table 1.

### 3.2 CONVERGENCE RESULTS

To derive the non-asymptotic convergence results of LV-HBA, we first demonstrate the decreasing property of a merit function introduced below,

$$V_k := \phi_{c_k}(x^k, y^k, z^k) + C_{\theta\lambda} \left\| (\theta^k, \lambda^k) - (\theta_r^*(x^k, y^k, z^k), \lambda_r^*(x^k, y^k, z^k)) \right\|^2, \tag{12}$$

where $C_{\theta\lambda} := \max\{(L_f + C_Z L_{g_1})^2 + 1/(2\gamma_1^2) + L_g^2, 1/\gamma_2^2\}$, $C_Z := \max_{z \in Z} \|z\|$, and

$$\phi_{c_k}(x, y, z) := \frac{1}{c_k} \big( F(x,y) - \underline{F} \big) + f(x,y) - v_{\gamma,r}(x,y,z). \tag{13}$$

**Lemma 3.1.** *Under Assumptions 3.1, 3.2 and 3.3, let $\gamma_1 \in (0, 1/\rho_f)$, $\gamma_2 > 0$, $c_{k+1} \geq c_k$ and $\eta_k \in (\underline{\eta}, \rho_T/L_B^2)$ with $\underline{\eta} > 0$, $\rho_T := \min\{1/\gamma_1 - \rho_f, 1/\gamma_2\}$ and $L_B := \max\{L_f + L_g + C_Z L_{g_2} + 1/\gamma_1, L_g + 1/\gamma_2\}$, then there exist constants $c_\alpha, c_\beta > 0$ such that when $\alpha_k \in (0, c_\alpha]$ and $\beta_k \in (0, c_\beta]$, the sequence of $(x^k, y^k, z^k)$ generated by LV-HBA satisfies*

$$\begin{aligned} V_{k+1} - V_k \leq & -\frac{1}{4\alpha_k} \|(x^{k+1}, y^{k+1}) - (x^k, y^k)\|^2 - \frac{1}{4\beta_k} \|z^{k+1} - z^k\|^2 \\ & - \underline{\eta} \rho_T C_{\theta\lambda} \left\| (\theta^k, \lambda^k) - (\theta_r^*(x^k, y^k, z^k), \lambda_r^*(x^k, y^k, z^k)) \right\|^2. \end{aligned} \tag{14}$$

The step sizes are carefully chosen to guarantee the sufficient descent property of $V_k$. This is essential for the non-asymptotic convergence analysis. Comprehensive proofs for Lemma 3.1 and accompanying auxiliary lemmas can be found in Sections A.4 , A.5 in Appendix.

Given the decreasing property of $V_k$, we proceed to establish the non-asymptotic convergence analysis. Owing to the constraint $f(x,y) - v_{\gamma,r}(x,y,z) \leq 0$ in (6), by employing an argument analogous to Ye & Zhu (1995), it can be deduced that conventional constraint qualifications are not satisfied

---

*Recall that a function $h$ is said to be $L$-smooth on $\Omega$ if $h$ is continuously differentiable and its gradient $\nabla h$ is $L$-Lipschitz continuous on $\Omega$.

at any feasible point of constrained problem (6). As a result, the standard KKT conditions are inappropriate as necessary optimality conditions for problem (6). Motivated by the approximate KKT condition presented in Andreani et al. (2010), which is characterized as an optimality condition for nonlinear program, regardless of constraint qualifications' fulfillment, we consider the following residual function $R_k := R_k(x, y, z)$ as a stationarity measure,

$$R_k := \text{dist}\left(0, (\nabla F(x, y), 0) + c_k\left((\nabla f(x, y), 0) - \nabla v_{\gamma, r}(x, y, z)\right) + \mathcal{N}_{C \times Z}(x, y, z)\right), \quad (15)$$

where $\mathcal{N}_\Omega(s)$ denotes the normal cone to $\Omega$ at $s$. This residual function $R_k(x, y, z)$ also serves as a stationarity measure for the penalized problem of (6), with $c_k$ serving as the penalty parameter,

$$\min_{(x, y) \in C, z \in Z} \psi_{c_k}(x, y, z) := F(x, y) + c_k\left(f(x, y) - v_{\gamma, r}(x, y, z)\right). \quad (16)$$

Evidently, $R_k(x, y, z) = 0$ if and only if $(x, y, z)$ is a stationary point for problem (16), meaning $0 \in \nabla \psi_{c_k}(x, y, z) + \mathcal{N}_{C \times Z}(x, y, z)$. The following theorem offers the non-asymptotic convergence for LV-HBA, and the proof is detailed in Section A.6 of the Appendix.

**Theorem 3.1.** *Under Assumptions of Lemma 3.1, let $\gamma_1 \in (0, 1/\rho_f)$, $\gamma_2 > 0$, $c_k = \underline{c}(k + 1)^p$ with $p \in (0, 1/2), \underline{c} > 0$ and $\eta_k \in (\underline{\eta}, \rho_T/L_B^2)$, then there exist $c_\alpha, c_\beta > 0$ such that when $\alpha_k \in (\underline{\alpha}, c_\alpha)$ and $\beta_k \in (\underline{\beta}, c_\beta)$ with $\underline{\alpha}, \underline{\beta} > 0$, the sequence of $(x^k, y^k, z^k, \theta^k, \lambda^k)$ generated by LV-HBA satisfies*

$$\min_{0 \leq k \leq K} \left\|(\theta^k, \lambda^k) - (\theta_r^*(x^k, y^k, z^k), \lambda_r^*(x^k, y^k, z^k))\right\| = O\left(\frac{1}{K^{1/2}}\right),$$

*and*

$$\min_{0 \leq k \leq K} R_k(x^{k+1}, y^{k+1}, z^{k+1}) = O\left(\frac{1}{K^{(1-2p)/2}}\right).$$

*Furthermore, if there exists $M > 0$ such that $\psi_{c_k}(x^k, y^k, z^k) \leq M$ for any $k$, the sequence of $(x^k, y^k, z^k)$ satisfies*

$$0 \leq f(x^K, y^K) - v_\gamma(x^K, y^K, z^K) \leq f(x^K, y^K) - v_{\gamma, r}(x^K, y^K, z^K) = O\left(\frac{1}{K^p}\right).$$

## 4 EXPERIMENTS

In this section, we evaluate the empirical performance of LV-HBA using numerical experiments on synthetic problems, hyperparameter optimization for SVM, and federated bilevel learning problem. We compare LV-HBA against AiPOD, E-AiPOD (Xiao et al., 2023b), and GAM (Xu & Zhu, 2023). The results underscore the effectiveness of LV-HBA in practical scenarios. Detailed experimental settings and parameter configurations can be found in Appendix A.1.The code is available at `https://github.com/SUSTech-Optimization/LV-HBA`.

### 4.1 SYNTHETIC EXPERIMENTS.

We test LV-HBA in comparison to AiPOD and E-AiPOD, on two synthetic coupling equality-constrained BLOs from two distinct scenarios: merely convex and strongly convex LL objectives.

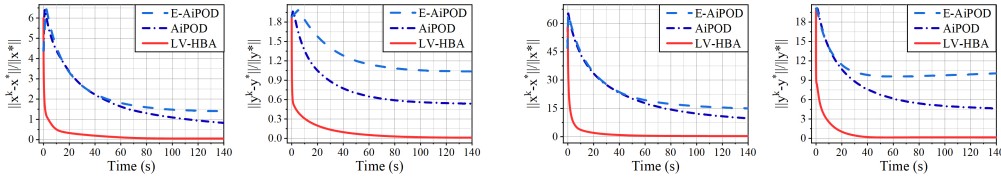

Figure 1: Comparison between AiPOD, E-AiPOD and LV-HBA on LL merely convex synthetic problem. Left two figures: initial point $10 \cdot \mathbf{1} \in \mathbb{R}^{300}$. Right two figures: initial point $100 \cdot \mathbf{1} \in \mathbb{R}^{300}$.

**LL merely convex.** We consider BLO with coupling equality constraints given by

$$\min_{x \in \mathbb{R}^n, y := (y_1, y_2) \in \mathbb{R}^{2n}} \frac{1}{2} \|x - y_2\|^2 + \frac{1}{2} \|y_1 - \mathbf{1}\|^2 \quad \text{s.t.} \quad y \in \arg\min_{y' \in \mathcal{Y}(x)} \left\{\frac{1}{2} \|y_1'\|^2 - x^T y_1' + \mathbf{1}^T y_2'\right\},$$

where $\mathbf{1} \in \mathbb{R}^n$ represents a vector with all elements equal to 1 and $\mathcal{Y}(x) = \{y \in \mathbb{R}^{2n} \mid \mathbf{1}^T x + \mathbf{1}^T y_1 + \mathbf{1}^T y_2 = 0\}$. Its optimal solution can be analytically expressed as $x^* = -\frac{3}{10}\mathbf{1}$, $y_1^* = \frac{7}{10}\mathbf{1}$, $y_2^* = -\frac{4}{10}\mathbf{1}$. For the problem where $n = 100$, we test algorithms from two distinct initial points: $10 \cdot \mathbf{1} \in \mathbb{R}^{300}$ and $100 \cdot \mathbf{1} \in \mathbb{R}^{300}$. The convergence curves relative to time are presented in Figure 1. Notably, LV-HBA provides a more precise approximation to the optimal solution and demonstrates faster convergence compared to AiPOD and E-AiPOD. The inadequate performance of AiPOD and E-AiPOD may stem from the fact that while our synthetic problem has a merely convex LL objective, both AiPOD and E-AiPOD require a strongly convex LL objective for convergence. Moreover, we examine the sensitivity of parameter $c_k$ in LV-HBA, and present its convergence curve in Figure 2. We further test the synthetic problem in a high-dimensional setting to demonstrate the computational efficiency of LV-HBA by increasing the dimension $n$. We record the time when $\|x^k - x^*\|/\|x^*\| \leq 10^{-2}$ is met by the iterates generated by LV-HBA. Results in Figure 2 highlight the computational efficiency of LV-HBA.

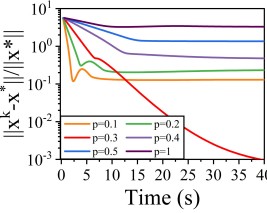 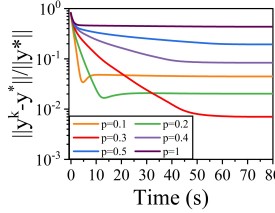 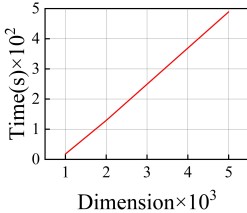

Figure 2: Left two figures: Impact of $p$ in parameter $c_k$ for LV-HBA. Rightmost figure: Time taken to achieve a specified accuracy v.s. dimension for LV-HBA.

**LL strongly convex.** We consider the strongly convex instance as presented in Xiao et al. (2023b):

$$\min_{x \in \mathcal{X}} \sin\left(c^\top x + d^\top y^*(x)\right) + \ln\left(\|x + y^*(x)\|^2 + 1\right) \quad \text{s.t.} \quad y^*(x) = \arg\min_{y \in \mathcal{Y}(x)} \frac{1}{2}\|x - y\|^2,$$

where $\mathcal{X} = \{x \mid Bx = 0\} \subset \mathbb{R}^{100}, \mathcal{Y}(x) = \{y \mid Ay + Hx = 0\} \subset \mathbb{R}^{100}$, and $A, B, H, c, d$ are non-zero matrices or vectors imported from the code of Xiao et al. (2023b) available at `https://github.com/hanshen95/AiPOD.`. Contrary to the experiment in Xiao et al. (2023b), we do not add Gaussian noise in this simulation. We test algorithms from two different initial points: $5 \cdot (\mathbf{1}, \mathbf{1})$ and $10 \cdot (\mathbf{1}, \mathbf{1})$ in $\mathbb{R}^{200}$. We use the norm of $\|y^k - y^*(x)\|$ as one stationarity measure, another one is the value of hyper-objective $F(x^k, y^*(x^k))$. We depict the respective convergence curves over time in Figure 3. Empirical results highlight the superior speed of convergence of our LV-HBA compared to both AiPOD and E-AiPOD.

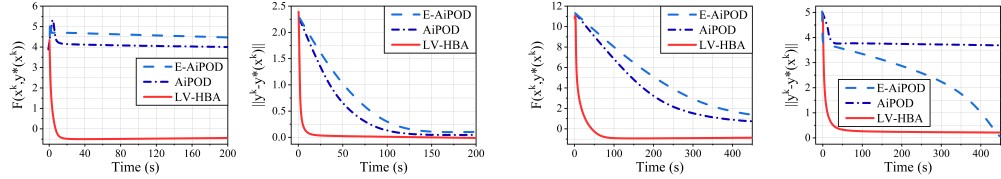

Figure 3: Comparison between AiPOD, E-AiPOD and LV-HBA on BLO with LL strongly convex objective. Left: initial point $5(\mathbf{1}, \mathbf{1})$ in $\mathbb{R}^{200}$. Right: initial point $10(\mathbf{1}, \mathbf{1})$ in $\mathbb{R}^{200}$.

Table 2: Numerical results on the hyperparameter optimization problem of SVM and the data hyper-cleaning task. The notation $a \pm b$ signifies a mean value $a$ with a standard deviation of $b$ over 40 trials.

| | Linear SVM | | | | Data Hyper-Cleaning | |
|---|---|---|---|---|---|---|
| Dataset | diabetes | | fourclass | | gisette | |
| Method | GAM | LV-HBA | GAM | LV-HBA | GAM | LV-HBA |
| Accuracy | $74 \pm 1.4$ | $75.07 \pm 1.6$ | $75.2 \pm 1.6$ | $75.4 \pm 1.2$ | $94.2 \pm 0.5$ | $94.6 \pm 0.4$ |
| Time(s) | $33.06 \pm 2.2$ | $\mathbf{6.65 \pm 1.3}$ | $30 \pm 2.2$ | $\mathbf{6 \pm 1.8}$ | $200 \pm 11.5$ | $\mathbf{100 \pm 11.5}$ |

## 4.2 HYPERPARAMETER OPTIMIZATION

We test the performance of our algorithm LV-HBA in comparison to GAM (Xu & Zhu, 2023). Both algorithms are applied to the hyperparameter optimization problem of SVM and the data hyper-cleaning task, as described in Xu & Zhu (2023). A comprehensive discussion of the problem formulation and the specific implementation settings can be found in Appendix A.1.

**Hyperparameter Optimization of SVM** We center our attention on the linear SVM model and conduct experiments on the dataset diabetes from Dua et al. (2017) and the dataset fourclass from Ho & Kleinberg (1996). The results are presented in Table 2. Moreover, Figure 4 illustrates the curve between test accuracy and time for the result on the dataset diabetes. Notably, our LV-HBA outperforms GAM by achieving superior accuracy within a significantly reduced time.

**Data Hyper-Cleaning** We compare our LV-HBA against GAM on the data hyper-cleaning task (Franceschi et al. (2017); Shaban et al. (2019)), utilizing dataset gisette (Guyon et al., 2004). Results are tabulated in Table 2. A performance curve for test accuracy against time is depicted in Figure 4. Remarkably, LV-HBA surpasses GAM, delivering enhanced test accuracy in a shorter time.

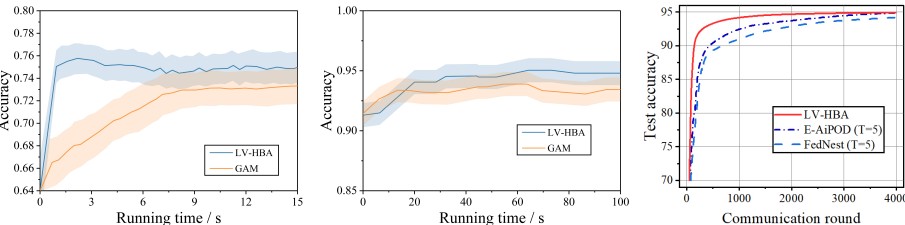

Figure 4: Left: accuracy v.s. running time in hyperparameter optimization of SVM on diabetes; Middle: accuracy v.s. running time in data hyper-cleaning; Right: accuracy v.s. communication round in federated loss function tuning problem.

## 4.3 FEDERATED LOSS FUNCTION TUNING

In this part, we test our LV-HBA compared to E-AiPOD (Xiao et al., 2023b) and FedNest (Tarzanagh et al., 2022) using the federated loss function tuning problem, as explored in Xiao et al. (2023b). Detailed descriptions of the problem formulation and the specific implementation settings are available in Appendix A.1. This federated learning with imbalanced data task aims to develop a model ensuring fairness and generalization across datasets dominated by under-represented classes Li et al. (2021). Results, detailed in Table 3 and Figure 4, indicate that LV-HBA surpasses E-AiPOD and FedNest in both communication complexity and computational efficiency.

Table 3: Communication round and time taken to achieve a specified accuracy in the experiment for the federated loss function tuning problem. E-A represents E-AiPOD; FN denotes FedNest; LV denotes LV-HBA.

| Test accuracy | Test accuracy: 90 | | | Test accuracy: 92 | | | Test accuracy: 94 | | |
|---|---|---|---|---|---|---|---|---|---|
| | E-A | FN | LV | E-A | FN | LV | E-A | FN | LV |
| Round | 527 | 625 | **65** | 1080 | 1575 | **149** | 2256 | 3010 | **588** |
| Time(s) | ×25.7 | ×27.4 | **1638** | ×23 | ×25.3 | **3754** | ×12 | ×15.9 | **14994** |

## 5 CONCLUSIONS

This work proposes a new approach and algorithm for solving a class of constrained BLO problems in which LL problem involves constraints coupling both UL and LL variables. The key enabling technique is to introduce a smooth proximal Lagrangian value function to handle the constrained LL problem. This allows us to smoothly reformulate the original BLO, and develop a Hessian-free gradient-based algorithm. In the future we would be interested in studying stochastic algorithms for constrained BLOs, leveraging the simplicity of our approach and incorporating techniques such as extrapolation, variance reduction, momentum, and others.

ACKNOWLEDGMENTS

Authors listed in alphabetical order. This work is supported by National Key R & D Program of China (2023YFA1011400), National Natural Science Foundation of China (12222106, 12326605, 62331014, 12371305), Guangdong Basic and Applied Basic Research Foundation (No. 2022B1515020082) and Shenzhen Science and Technology Program (No. RCYX20200714114700072).

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

## A  APPENDIX

The appendix is organized as follows:

- The experimental details is provided in Section A.1.
- Expanded related work is provided in Section A.2.
- The equivalent results of the reformulated problem 4 are provided in Section A.3.
- Some useful auxiliary lemmas are provided in Section A.4.
- The proof of Lemma 3.1 is given in Section A.5.
- The proof of Proposition 3.1 is provided in Section A.6.

### A.1  EXPERIMENTAL DETAILS

In this section, we outline the specific experimental settings. All experiments were conducted using Python 3.8 on a computer with an Intel(R) Xeon(R) Gold 5218R CPU @ 2.10GHz CPU and an NVIDIA A100 GPU with 40GB memory GPU.

#### A.1.1  SYNTHETIC EXPERIMENTS

**LL merely convex:**

Hyper-parameter settings for algorithms.

LV-HBA: In Figure 1, the step sizes are chosen as $\alpha = 0.005$, $\beta = 0.002$, $\eta = 0.03$, $\gamma_1 = \gamma_2 = 10$, $r = 1$ with parameter $c_k = (k+1)^{0.3}$. In Figure 2, the step sizes are chosen as $\alpha = 0.002$, $\beta = 0.002$, $\eta = 0.03$, $\gamma_1 = \gamma_2 = 0.1$, $r = 1$, $c_k = (k+1)^p$ with various $p$.

E-AiPOD: Projection probability is $p = 0.3$, total iterations are $K = 300/p$, UL iterations are $T = 2$, and LL iterations are $S = 5$. In Figure 1, the step sizes are set as $\alpha = 0.0001$, $\beta = 0.001$. In both AiPOD and E-AiPOD, the parameter $T$ is set to 2, as this choice has been demonstrated to yield best performance, as shown in (Xiao et al., 2023b, Figure 1).

**LL strongly convex Case:**

The strongly convex instance is adapted from Xiao et al. (2023b) by omitting the Gaussian noise.

Hyper-parameter settings for algorithms. For LV-HBA, the step sizes are chosen as $\alpha = 0.02$, $\beta = 0.001$, $\eta = 0.1$, $\gamma_1 = \gamma_2 = 1$, $r = 1000$ with parameter $c_k = (k+1)^{0.3}$. For both AiPOD and E-AiPOD, the hyper-parameters are set consistent with the code in Xiao et al. (2023b). Specifically, the projection probability $p = 0.3$, the number of UL iterations in is $T = 2$, the number of LL iterations is $S = 5$, and the step sizes are set as $\alpha = 0.001$, $\beta = 0.02$.

**Sensitivity of parameters:**

Additionally, to assess the sensitivity of the remaining parameters, further numerical experiments were conducted on LL merely convex synthetic model. We record the time when $\|x^k - x^*\|/\|x^*\| \leq 10^{-2}$ is met by the iterates generated by LV-HBA. The results are as follows:

| Parameters / Sensitivity | $\alpha$ | $\beta$ | $\eta$ | $\gamma = \gamma_1 = \gamma_2$ | $\underline{c}$ | $p$ | $r$ | Time(s) |
|---|---|---|---|---|---|---|---|---|
| $\alpha$ | 0.001 | 0.02 | 0.01 | 10 | 0.025 | 0.3 | 1000 | 0.23 |
| | 0.005 | 0.02 | 0.01 | 10 | 0.025 | 0.3 | 1000 | 0.064 |
| | 0.01 | 0.02 | 0.01 | 10 | 0.025 | 0.3 | 1000 | 0.052 |
| $\beta$ | 0.005 | 0.005 | 0.01 | 10 | 0.025 | 0.3 | 1000 | 0.068 |
| | 0.005 | 0.02 | 0.01 | 10 | 0.025 | 0.3 | 1000 | 0.064 |
| | 0.005 | 0.1 | 0.01 | 10 | 0.025 | 0.3 | 1000 | 0.063 |
| $\eta$ | 0.005 | 0.02 | 0.005 | 10 | 0.025 | 0.3 | 1000 | 0.098 |
| | 0.005 | 0.02 | 0.01 | 10 | 0.025 | 0.3 | 1000 | 0.064 |
| | 0.005 | 0.02 | 0.05 | 10 | 0.025 | 0.3 | 1000 | 0.025 |
| $\gamma = \gamma_1 = \gamma_2$ | 0.005 | 0.02 | 0.01 | 5 | 0.025 | 0.3 | 1000 | 0.069 |
| | 0.005 | 0.02 | 0.01 | 10 | 0.025 | 0.3 | 1000 | 0.064 |
| | 0.005 | 0.02 | 0.01 | 500 | 0.005 | 0.3 | 1000 | 0.064 |
| $\underline{c}$ | 0.005 | 0.02 | 0.01 | 10 | 0.005 | 0.3 | 1000 | 0.026 |
| | 0.005 | 0.02 | 0.01 | 10 | 0.05 | 0.3 | 1000 | 0.064 |
| | 0.005 | 0.02 | 0.01 | 10 | 0.025 | 0.3 | 1000 | 0.086 |
| $r$ | 0.005 | 0.02 | 0.01 | 10 | 0.025 | 0.3 | 200 | 0.066 |
| | 0.005 | 0.02 | 0.01 | 10 | 0.025 | 0.3 | 1000 | 0.064 |
| | 0.005 | 0.02 | 0.01 | 10 | 0.025 | 0.3 | 2000 | 0.065 |

### A.1.2 HYPERPAMETER OPTIMIZATION

The hyperparameter optimization of SVM and the data hyper-cleaning experiments were performed using qpth version 0.0.11 and cvxpy version 1.2.0.

**Hyperparameter Optimization of SVM** We test the performance of our algorithm LV-HBA in comparison to GAM proposed in Xu & Zhu (2023) on the same hyperparameter optimization problem of SVM as considered in Xu & Zhu (2023). Experiments are conducted using datasets diabetes from Dua et al. (2017). and fourclass from Ho & Kleinberg (1996). For dataset diabetes, we randomly partition it into training, validation, and testing subsets containing 500, 150, and 118 examples, respectively. Similarly, dataset fourclass is partitioned into training, validation, and testing subsets with 500, 150, and 212 examples, respectively. We conduct experiments on each dataset with 40 repetitions.

The hyperparameter optimization of SVM can be expressed as:

$$\min_c \Phi(c) = \mathcal{L}_{\mathcal{D}_{\text{val}}}\left(w^*, b^*\right),$$

where the hyperparameter to be optimized is $c = [c_1, \ldots, c_N]$ and $w^*, b^*$ are solution to the SVM optimization problem given by

$$(w^*, b^*, \xi^*) = \arg\min_{w, b, \xi} \frac{1}{2}\|w\|^2 + \frac{1}{2}\sum_{i=1}^{N} e^{c_i}\xi_i^2$$

$$\text{s.t. } l_i\left(w^\top \phi\left(z_i\right) + b\right) \geq 1 - \xi_i, \ i = 1, 2, \ldots N.$$

Here, $\mathcal{D}_{\text{val}}$ represents the validation data, and $\mathcal{D}_{\text{tr}}$ the training data. For all $1 \leq i \leq N$, $z_i$ denotes the data point, $l_i$ is the label, and $(z_i, l_i) \in \mathcal{D}_{\text{tr}}$. The upper-level objective function is:

$$\mathcal{L}_{\mathcal{D}_{\text{val}}}\left(w^*, b^*\right) = \frac{1}{|\mathcal{D}_{\text{val}}|} \sum_{(z,l) \in \mathcal{D}_{\text{val}}} \mathcal{L}\left(w^*, b^*; z, l\right),$$

where $\mathcal{L}\left(w^*, b^*; \mathcal{D}_{\text{val}}\right)$ is given by

$$\mathcal{L}\left(w^*, b^*; z, l\right) = \sigma\left(\frac{-l\left(z^\top w^* + b\right)}{\|w^*\|}\right),$$

with $\sigma(x) = \frac{1-e^{-x}}{1+e^{-x}}$. The term $\frac{l(z^\top w^* + b)}{\|w^*\|}$ signifies the signed distance between point $z$ and the decision plane $z^\top w^* + b = 0$. It is positive when predictions are accurate, and negative otherwise. Thus, $\mathcal{L}_{\mathcal{D}_{\text{val}}}(w^*, b^*)$ serves as a differentiable surrogate for validation accuracy.

Hyper-parameter settings for algorithms. For LV-HBA, the step sizes are chosen as $\alpha = 0.01$, $\beta = 0.1$, $\eta = 0.01$, $\gamma_1 = \gamma_2 = 10$, $r = 100$ with parameter $c_k = (k+1)^{0.3}$. For GAM, hyperparameters are set in alignment with the code in Xu & Zhu (2023): $\gamma = 0.3$, $\epsilon_0 = 0.3$, $\beta = 0.5$. In GAM's implementation, to uphold the LL strong convexity assumption, the LL problem's objective function is set as $\frac{1}{2}\|w\|^2 + \frac{1}{2}\sum_{i=1}^{N} e^{c_i}\xi_i^2 + \frac{1}{2}\mu b^2$ where $\mu$ is a small positive number.

**Data Hyper-Cleaning** We adopt the data hyper-cleaning formulation as the hyperparameter optimization of SVM, as presented in Xu & Zhu (2023). In this model, post-optimization of hyperparameter $c$, the penalty term corresponding to the corrupted data $(z_i, y_i)$ approaches 0. Consequently, the corrupted data $(z_i, y_i)$ is identified and has a negligible impact on the training and prediction of the classifier model. Experiments are conducted using the datasets gisette from Guyon et al. (2004). For dataset gisette, we segment it into training, validation, and testing subsets, comprising 400, 180, and 5420 examples, respectively. We conduct experiments on each dataset with 40 repetitions.

Hyper-parameter settings for algorithms. For LV-HBA, we select step sizes with values $\alpha = 0.01$, $\beta = 0.1$, and $\eta = 0.01$, accompanied by parameter $c_k = (k+1)^{0.3}$. For GAM, hyperparameters are consistent with specifications in Xu & Zhu (2023), with $\gamma = 0.3$, $\epsilon_0 = 0.3$, and $\beta = 0.5$.

### A.1.3 FEDERATED LOSS FUNCTION TUNING

In this part, we test our LV-HBA compared to E-AiPOD Xiao et al. (2023b) and FedNest Tarzanagh et al. (2022) on the same federated loss function tuning problem, as explored in Xiao et al. (2023b). The experiments were executed with opencv-python version 4.6.0.66. In the federated loss function tuning problem, the UL optimizes loss-tuning parameters to enhance both generalization and fairness. Meanwhile, the LL focuses on training model parameters on potentially imbalanced datasets. The formal problem statement is:

$$\min_{x \in \mathcal{X}} \frac{1}{M} \sum_{m=1}^{M} f_{\text{vs}}^{\text{up}}\left(y_m^*(x); \mathcal{D}_{\text{val}}^m\right),$$

$$\text{s.t. } y^*(x) = \arg\min_{y \in \mathcal{Y}} \frac{1}{M} \sum_{m=1}^{M} f_{\text{vs}}^{\text{low}}\left(x, y_m; \mathcal{D}_{\text{tr}}^m\right),$$

where $M = 50$ representing the number of clients, $x$ is the loss-tuning parameter and $y$ indicates the neural network parameters. $\mathcal{D}_{\text{tr}}^m$ and $\mathcal{D}_{\text{val}}^m$ are the training and validation sets of client $m$. The consensus sets $\mathcal{X}$ and $\mathcal{Y}$ are given by $\mathcal{X} := \{x \mid x_1 = \cdots = x_M\}$ and $\mathcal{Y} := \{y \mid y_1 = \cdots = y_M\}$, respectively. Training datasets $\{\mathcal{D}_{\text{tr}}^m\}_{m=1}^{M}$ have class imbalances. As introduced by Kini et al. (2021), the vector-scaling loss $f_{\text{vs}}^{\text{low}}$ is

$$f_{\text{vs}}^{\text{low}}(x, y; \mathcal{D}) := -\frac{1}{|\mathcal{D}|} \sum_{d_n \in \mathcal{D}} \omega_{l_n} \log \frac{\exp\left(\delta_{l_n} h_{l_n}(y; d_n) + \tau_{l_n}\right)}{\sum_{c=1}^{C} \exp\left(\delta_c h_c(y; d_n) + \tau_c\right)},$$

where $N$ signifies dataset size, $C$ denotes the class count, and $d_n$ is the $n$-th data item with label $l_n$ in dataset $\mathcal{D}$. The logit output of the neural network for parameters $y$ and input $d_n$ is $h(y; d_n) = [h_1(y; d_n), \ldots, h_C(y; d_n)]^\top \in \mathbb{R}^C$, $x$ is defined as $(\omega, \delta, \tau)$ with $\omega := [\omega_1, \ldots, \omega_C]^\top \in \mathbb{R}^C$, and $\delta, \tau$ defined in a similar manner. The upper-level loss $f_{\text{vs}}^{\text{up}}$ is a variant of $f_{\text{vs}}^{\text{low}}$ where $\delta = 1$, $\tau = 0$, and $\omega$ is a static class weight for the validation dataset.

Hyper-parameter settings for algorithms. For LV-HBA, we select step sizes with values $\alpha = 0.01$, $\beta = 0.01$, and $\eta = 0.01$, $\gamma_1 = \gamma_2 = 10$, $r = 100$, parameter $c_k = (k+1)^{0.3}$ and bath size as 256. For E-AiPOD and FedNest, hyperparameters align with specifications in Xiao et al. (2023b): E-AiPOD has a communication probability $p = 0.3$, $S = 20$, $\alpha = 0.01$, $\beta = 0.04$, $N' = 3$, and batch size of 256. FedNest utilizes LL iteration number $\tau = 3$, episode $T = 3$, resulting in a communication frequency of 0.3 per LL iteration.

## A.2 EXPANDED RELATED WORK

In this section, we provide an extensive review of recent studies closely related to our work.

**Approaches for BLO.** One of the most commonly employed approaches for tackling BLO problems is to reformulate them as single-level problems. This can often be accomplished in two ways Dempe & Zemkoho (2013). One of these approaches, known as the KKT (or MPEC) reformulation, replaces the LL problem with its Karush-Kuhn-Tucker (KKT) conditions and minimizes over the original variables as well as multipliers if the LL constraints exist. The resulting problem is the so-called mathematical program with complementarity/equilibrium constraints (MPCC/MPEC) Luo et al. (1996), which itself poses a significant challenge when treated as a nonlinear programming problem Kim et al. (2020). Remarkably, the KKT reformulation employs the KKT conditions, thereby unavoidably relying on first-order gradient information. Consequently, gradient-based algorithms based on KKT reformulation also depend on second-order gradient information.

Another often used approach is the value function approach, originally proposed in Outrata (1990) and Ye & Zhu (1995). It is obtained by replacing the LL problem by its description via the (optimal) value function. Unlike the KKT reformulation, the value function reformation does not use any gradient information of the objective and constraint functions in the LL problem. To the best of our knowledge, the majority of existing Hessian-free (also referred to as fully first-order) gradient-based algorithms for both unconstrained and constrained BLOs, are developed based on the value function reformulation, see, e.g., Liu et al. (2021b; 2023a); Ye et al. (2022); Sow et al. (2022); Shen & Chen (2023); Kwon et al. (2023a); Lu & Mei (2023). It should be noted, however, that the value function is typically nonsmooth, even when the functions involved are linear and affine. Hence, the value function reformulation often leads to a nonsmooth problem. To alleviate the nonsmooth issue, the recent works Ye et al. (2023); Gao et al. (2022) develop difference of convex algorithms for solving BLO problems in which the UL objective is a difference of convex function and the LL problem is fully convex.

Recently, to weaken the underlying assumption from lower level full convexity to weak convexity, Gao et al. (2023) proposes a new reformulation of BLOs, using Moreau envelope of the LL problem. They also demonstrate the equivalence between the reformulated and the original BLO problems in the convex setting. Other approaches for BLOs include implicit methods Franceschi et al. (2017); Ghadimi & Wang (2018); Shaban et al. (2019), penalty methods Lin et al. (2014), duality-based solution approach Ouattara & Aswani (2018); Li et al. (2023; 2024).

**Unconstrained BLO.** The LL strong convexity in unconstrained BLO significantly contributes to the development of efficient BLO algorithms, see, e.g., Maclaurin et al. (2015); Franceschi et al. (2017); Shaban et al. (2019); Mackay et al. (2019); Grazzi et al. (2020); Ji et al. (2021; 2022) for the iterative differentiation (ITD) based approach; Pedregosa (2016); Ghadimi & Wang (2018); Rajeswaran et al. (2019); Lorraine et al. (2020); Hong et al. (2023); Chen et al. (2021); Arbel & Mairal (2022a); Dagréou et al. (2022); Ye et al. (2022); Yang et al. (2023a) for the approximate implicit differentiation (AID) based approach. Recently, based on the value function-based reformulation, Kwon et al. (2023a) developed stochastic and deterministic fully first-order BLO algorithms and established their non-asymptotic convergence guarantees, while an improved convergence analysis is provided in the recent work Chen et al. (2023a).

Convex LL problems introduce additional challenges, such as the presence of multiple LL solutions (Non-Singleton), which can impede the utilization of implicit-based approaches developed for nonconvex-strongly-convex BLO. To tackle Non-Singleton, recent advances include: aggregation methods (or called sequential averaging methods) in Liu et al. (2020); Li et al. (2020); Liu et al. (2022) with asymptotic convergence guarantees; in Liu et al. (2023b) with convergence rate analysis; value function-based difference-of-convex algorithm in Ye et al. (2023); Gao et al. (2022); primal-dual algorithms in Sow et al. (2022); min-max optimization reformulation-based first-order penalty methods in Lu & Mei (2023).

Efficient methods for nonconvex-nonconvex BLO remain under-explored, recent advances include: initialization auxiliary and pessimistic trajectory truncation method in Liu et al. (2021c); value function-based interior-point method in Liu et al. (2021b); possibly degenerate implicit differentiation-based unrolled optimization algorithms in Arbel & Mairal (2022b); momentum-based algorithm in Huang (2023); generalized alternating method in Xiao et al. (2023a); fully first-

order value function-based algorithm in Ye et al. (2022); penalty-based fully first-order algorithm in Shen & Chen (2023); a smoothed first-order Lagrangian method in Lu (2024).

**Constrained BLO.** While gradient-based algorithms for unconstrained BLO problems have been extensively explored, the investigation of efficient methods for constrained BLO problems is relatively limited, especially when addressing LL constraints coupling both UL and LL variables.

Recently, driven by applications in machine learning, the recent works such as Tsaknakis et al. (2022); Khanduri et al. (2023) study the constrained BLOs, where the LL problem involves the minimization of a strongly convex objective over a set of linear inequality constraints; Xiao et al. (2023a) investigates the stochastic BLO problems with possibly coupled equality constraints in both upper and lower levels; Xu & Zhu (2023) considers BLOs in which the LL problem is convex with general equality and inequality constraints, while assuming that the LL objective is strongly convex and the constraints satisfy strict Linear Independence Constraint Qualification (LICQ); Tsaknakis et al. (2023) develop a novel barrier-based gradient approximation algorithm that transforms the general constrained BLO problem to a problem with only linear equality constraints. Observe that all of these works employ implicit gradient-based methods, relying on the computation of the implicit gradient of the unique LL solution mapping. Among them, Khanduri et al. (2023) develops a linear perturbation-based smoothing framework for the linearly constrained LL problem that ensures the existence of the implicit gradient in an almost sure sense. Notably, these implicit gradient-based methods for constrained BLOs unavoidably rely on computationally intensive calculations related to the Hessian matrix.

The value function-based methods can avoid recurrent calculations related to the Hessian matrix, see, e.g., Liu et al. (2023a); Lu & Mei (2023). Both papers have considered BLOs with general constraints in the LL problem. For constrained BLOs, value function-based methods face challenges related to non-differentiable constraints, stemming from the value function-based reformulation. To address the nonsmooth issue, Liu et al. (2023a) proposes a sequential minimization algorithmic framework, by adding a quadratic regularization and penalty/barrier functions of the LL inequality constraints to the LL objective; Lu & Mei (2023) develops first-order penalty methods by solving a sequence of minimax problems or a single minimax problem. Recent advances include primal-dual algorithms in Sow et al. (2022); primal nonsmooth reformulation-based algorithm in Helou et al. (2023); penalty-based first-order algorithms in Kwon et al. (2023b).

There is also a line of works devoted to tackle the constrained UL setting including: implicit approximation methods in Ghadimi & Wang (2018); a two-timescale framework in Hong et al. (2023); a single-timescale method in Chen et al. (2022a); initialization auxiliary method in Liu et al. (2021c); Bregman distance-based method in Huang et al. (2022); value function-based Difference-of-Convex algorithm in Gao et al. (2022); proximal gradient-type algorithm in Chen et al. (2022b); penalty-based method in Shen & Chen (2023); Moreau Envelope-based Difference-of-weakly-Convex method in Gao et al. (2023); inexact conditional gradient method in Abolfazli et al. (2023); nested compositional BLO in Chen et al. (2023b).

## A.3 EQUIVALENT RESULTS OF REFORMULATED PROBLEM

In the following result, we demonstrate that the reformulation problem (4) is equivalent to the BLO problem (1).

Recall that for the sake of notational simplicity, throughout this paper, for LL constraint mapping $g : X \times Y \to \mathbb{R}^l$ and any vectors $\lambda, z \in \mathbb{R}^l$, we represent $\sum_{i=1}^l \lambda_i g_i$ and $\sum_{i=1}^l z_i g_i$ as $\lambda g$ and $zg$, respectively. Similarly, $\sum_{i=1}^l \lambda_i \nabla_y g_i$ and $\sum_{i=1}^l z_i \nabla_y g_i$ are denoted by $\lambda \nabla_y g$ and $z \nabla_y g$, respectively.

For the reader's convenience, we restate the reformulation problem (4) as follows:

$$\min_{(x,y)\in C, z\geq 0} F(x,y) \quad \text{s.t.} \quad f(x,y) - v_\gamma(x,y,z) \leq 0, \tag{4}$$

where $C := \{(x,y) \in X \times Y \mid g(x,y) \leq 0\}$, and $v_\gamma(x,y,z)$ is the proximal Lagrangian value function, restated below,

$$v_\gamma(x,y,z) := \min_{\theta \in Y} \max_{\lambda \in \mathbb{R}_+^l} \left\{ f(x,\theta) + \lambda g(x,\theta) + \frac{1}{2\gamma_1}\|\theta - y\|^2 - \frac{1}{2\gamma_2}\|\lambda - z\|^2 \right\}. \tag{3}$$

The proximal Lagrangian value function $v_\gamma(x, y, z)$ defined in equation (3) can be regarded as the (upper) Yosida approximate of the Lagrangian function $\mathcal{L}(x, y, \lambda) := f(x, y) + \lambda g(x, y)$ of the LL problem (Attouch & Wets, 1983, Section 5).

**Theorem A.1.** *Assume that $f(x, \cdot)$ and $g(x, \cdot)$ are both convex on $Y$. Suppose $\gamma_1, \gamma_2 > 0$, the reformulated problem (4) is equivalent to the BLO problem (1), if multiplier of the lower-level problem( 2) exists for any feasible point $(x, y)$ of the BLO problem (1).*

*Proof.* First, let $(x, y, z)$ be any feasible point of problem (4), then we have $(x, y) \in X \times Y$, $z \geq 0$, and $g(x, y) \leq 0$. Furthermore, the following inequalities hold,

$$
\begin{aligned}
f(x, y) \leq v_\gamma(x, y, z) &:= \min_{\theta \in Y} \max_{\lambda \geq 0} \left\{ f(x, \theta) + \lambda g(x, \theta) + \frac{1}{2\gamma_1} \|\theta - y\|^2 - \frac{1}{2\gamma_2} \|\lambda - z\|^2 \right\} \\
&\leq \min_{\theta \in Y} \max_{\lambda \geq 0} \left\{ f(x, \theta) + \lambda g(x, \theta) + \frac{1}{2\gamma_1} \|\theta - y\|^2 \right\} \\
&\leq \min_{\theta \in Y} \left\{ f(x, \theta) + \frac{1}{2\gamma_1} \|\theta - y\|^2 \,\big|\, g(x, \theta) \leq 0 \right\} \\
&\leq f(x, y).
\end{aligned}
\tag{17}
$$

Since the first and the last terms in the above inequalities are the same, we must have equalities throughout. Specially, we have

$$
y \in \operatorname*{argmin}_{\theta \in Y} \left\{ f(x, \theta) + \frac{1}{2\gamma_1} \|\theta - y\|^2 \,\big|\, g(x, \theta) \leq 0 \right\}.
$$

Because $f(x, \cdot)$ and $g(x, \cdot)$ are both convex functions on $Y$, considering $g(x, \cdot) \leq 0$ as an abstract convex set constraint and using the first-order optimality conditions, we obtain that

$$
y \in \operatorname{argmin}_{\theta \in Y} \left\{ f(x, \theta) \,|\, g(x, \theta) \leq 0 \right\},
$$

and thus $y \in S(x)$. Then the point $(x, y)$ is feasible to the BLO problem (1).

Conversely, suppose that $(x, y)$ is an feasible point of the BLO problem (1), then we have $(x, y) \in X \times Y$ and $y \in S(x)$. On one hand, according to the assumption that multiplier $z \geq 0$ of the LL problem (2) exists at $(x, y)$, since the LL problem is convex, by the first-order optimality conditions, we get

$$
y \in \operatorname{argmin}_{\theta \in Y} \{ f(x, \theta) + z g(x, \theta) \}.
$$

Once more, due to the convexity of the LL problem, this implies that

$$
y \in \operatorname{argmin}_{\theta \in Y} \left\{ f(x, \theta) + z g(x, \theta) + \frac{1}{2\gamma_1} \|\theta - y\|^2 \right\}.
$$

On the other hand, since $g(x, y) \leq 0$, by the complementarity conditions, i.e., $z g(x, y) = 0$,

$$
z \in \operatorname{argmax}_{\lambda \geq 0} \left\{ f(x, y) + \lambda g(x, y) - \frac{1}{2\gamma_2} \|\lambda - z\|^2 \right\}.
$$

Hence, the point $(y, z)$ is a saddle point of the following strong convex strong concave function $L : Y \times \mathbb{R}_+^l \to \mathbb{R}$:

$$
L(\theta, \lambda) := f(x, \theta) + \lambda g(x, \theta) + \frac{1}{2\gamma_1} \|\theta - y\|^2 - \frac{1}{2\gamma_2} \|\lambda - z\|^2.
$$

Thus it follows from saddle point theorem that

$$
\begin{aligned}
&\min_{\theta \in Y} \max_{\lambda \geq 0} \left\{ f(x, \theta) + \lambda g(x, \theta) + \frac{1}{2\gamma_1} \|\theta - y\|^2 - \frac{1}{2\gamma_2} \|\lambda - z\|^2 \right\} \\
&= f(x, y) + z g(x, y) = f(x, y),
\end{aligned}
$$

where the last inequality uses the complementarity conditions $z g(x, y) = 0$. Therefore, $v_\gamma(x, y, z) = f(x, y)$ and then $(x, y, z)$ is feasible to the reformulation problem (4). $\qquad \square$

**Remark A.1.** *Indeed, as per the estimate provided in (17), the proximal Lagrangian value function $v_\gamma(x, y, z)$ establishes a lower bound for $f(x, y)$, that is, $v_\gamma(x, y, z) \leq f(x, y)$ for any $(x, y, z) \in C \times \mathbb{R}^l_+$. Specifically, we have*

$$v_\gamma(x, y, z) \leq \min_{\theta \in Y} \left\{ f(x, \theta) + \frac{1}{2\gamma_1} \|\theta - y\|^2 \mid g(x, \theta) \leq 0 \right\} =: v_{\gamma_1}(x, y), \tag{18}$$

*where $v_{\gamma_1}(x, y)$ coincides with the so-called Moreau envelope function, initially introduced by Gao et al. (2023) for BLO problems. Additionally, the latter also acts as a lower bound for the function $f(x, y)$.*

Subsequently, we demonstrate that for a sufficiently large $r$, the solution to the reformulation (4) can be obtained by solving variant (6). Note that $\text{Proj}_Z$ in Algorithm 1 is a simple Euclidean projection on a box $Z := [0, r]^l$ with lower bounds 0 and upper bounds $r \geq 0$. This projection is pivotal in guaranteeing the boundedness of the auxiliary variable $z^{k+1}$. For clarity, we restate the variant (6) of the reformulation (4) as follows:

$$\min_{(x,y) \in X \times Y, z \in Z} F(x, y) \quad \text{s.t.} \quad f(x, y) - v_{\gamma,r}(x, y, z) \leq 0, \quad g(x, y) \leq 0, \tag{6}$$

where $v_{\gamma,r}(x, y, z)$ is a truncated proximal Lagrangian value function, defined in equation (5),

$$v_{\gamma,r}(x, y, z) := \min_{\theta \in Y} \max_{\lambda \in Z} \left\{ f(x, \theta) + \lambda g(x, \theta) + \frac{1}{2\gamma_1} \|\theta - y\|^2 - \frac{1}{2\gamma_2} \|\lambda - z\|^2 \right\}. \tag{5}$$

**Theorem A.2.** *Suppose $\gamma_1, \gamma_2 > 0$, let an optimal solution $(x^*, y^*, z^*)$ of reformulation (4) exist such that $z^*$ is within the set $Z$. Then $(x^*, y^*, z^*)$ is also an optimal solution for the reformulation (6). Consequently, the optimal values for both reformulations, (4) and (6), are identical. Moreover, any optimal solution of (6) is optimal to reformulation (4).*

*Proof.* Firstly, according to the definitions of $v_\gamma$ and $v_{\gamma,r}$, we have that $v_\gamma(x, y, z) \geq v_{\gamma,r}(x, y, z)$ for any $(x, y, z) \in X \times Y \times Z$. Therefore, any feasible point $(x, y, z)$ of problem (6) is also feasible to problem (4) and thus the optimal value of problem (6) is larger or equal to that of problem (4).

Conversely, let $(x^*, y^*, z^*)$ be an optimal solution of the reformulation problem (4) with $z^*$ belonging to the set $Z$, since $(x^*, y^*, z^*)$ is feasible to problem (4). As shown in the proof of Theorem A.1, we obtain $y^* \in S(x^*)$. Next we show that $z^*$ is a multiplier of the LL problem (2) at $(x^*, y^*)$. To this end, it suffices to prove that

$$(y^*, z^*) = \operatorname*{argmin}_{\theta \in Y} \operatorname*{argmax}_{\lambda \geq 0} \left\{ f(x^*, \theta) + \lambda g(x^*, \theta) + \frac{1}{2\gamma_1} \|\theta - y^*\|^2 - \frac{1}{2\gamma_2} \|\lambda - z^*\|^2 \right\}. \tag{19}$$

By estimates in (17), we get

$$y^* \in \operatorname*{argmin}_{\theta \in Y} \max_{\lambda \geq 0} \left\{ f(x^*, \theta) + \lambda g(x^*, \theta) + \frac{1}{2\gamma_1} \|\theta - y^*\|^2 \right\},$$

$$y^* = \operatorname*{argmin}_{\theta \in Y} \max_{\lambda \geq 0} \left\{ f(x^*, \theta) + \lambda g(x^*, \theta) + \frac{1}{2\gamma_1} \|\theta - y^*\|^2 - \frac{1}{2\gamma_2} \|\lambda - z^*\|^2 \right\}.$$

Furthermore, the optimal values of the above optimization problems are equal. This implies that $z^* g(x^*, y^*) = 0$, and then

$$z^* \in \operatorname*{argmax}_{\lambda \geq 0} \left\{ f(x^*, y^*) + \lambda g(x^*, y^*) - \frac{1}{2\gamma_2} \|\lambda - z^*\|^2 \right\}.$$

Now since $z^* \in Z$, we have

$$(y^*, z^*) = \operatorname*{argmin}_{\theta \in Y} \operatorname*{argmax}_{\lambda \in Z} \left\{ f(x^*, \theta) + \lambda g(x^*, \theta) + \frac{1}{2\gamma_1} \|\theta - y^*\|^2 - \frac{1}{2\gamma_2} \|\lambda - z^*\|^2 \right\},$$

leading to $v_\gamma(x^*, y^*, z^*) = v_{\gamma,r}(x^*, y^*, z^*)$ and thus $(x^*, y^*, z^*)$ is also feasible to problem (6). Therefore, the optimal value of problem (6) is equal to that of problem (4). Then, because any feasible point $(x, y, z)$ of problem (6) is feasible to problem (4), we get the conclusion. $\square$

A.4 AUXILIARY LEMMAS

The following lemma provides a characterization of the gradient of the Lagrangian based proximal value function $v_{\gamma,r}(x, y, z)$.

Given that $f$ is $L_f$-smooth on $X \times Y$, by leveraging the descent lemma (Beck, 2017, Lemma 5.7), it can be deduced that $f$ is also $L_f$-weakly convex, i.e, $f(x, y) + L_f \|(x, y)\|^2/2$ is convex on $X \times Y$. Consequently, under Assumption 3.2, $f$ is $\rho_f$-weakly convex on $X \times Y$, with $\rho_f \geq 0$ potentially being smaller than $L_f$. To precisely determine the range for the step sizes of LV-HBA , we will employ the weak convexity constant of $f$, $\rho_f$, in subsequent results.

**Lemma A.1.** *Under Assumptions 3.2 and 3.3, let $\gamma_1 \in (0, 1/\rho_f)$ and $\gamma_2 > 0$. Then $v_{\gamma,r}(x, y, z)$ is continuously differentiable on $X \times Y \times \mathbb{R}^l$, and for any $(x, y, z) \in X \times Y \times \mathbb{R}^l$,*

$$\nabla v_{\gamma,r}(x, y, z) = \left( \nabla_x f(x, \theta^*) + \lambda^* \nabla_x g(x, \theta^*), \frac{(y - \theta^*)}{\gamma_1}, \frac{(\lambda^* - z)}{\gamma_2} \right), \tag{20}$$

*where $\theta^* := \theta^*(x, y, z)$ and $\lambda^* := \lambda^*(x, y, z)$ is the unique saddle point of the min-max problem:*

$$\min_{\theta \in Y} \max_{\lambda \in Z} \left\{ f(x, \theta) + \lambda g(x, \theta) + \frac{1}{2\gamma_1} \|\theta - y\|^2 - \frac{1}{2\gamma_2} \|\lambda - z\|^2 \right\}. \tag{21}$$

*Furthermore, for any $\rho_v \geq \rho_f/(1 - \gamma_1 \rho_f)$, $v_{\gamma,r}(x, y, z)$ is $\rho_v$-weakly convex with respect to variables $(x, y)$ on $X \times Y$ for any fixed $z \in \mathbb{R}^l$.*

*Proof.* Firstly, we define an auxiliary function,

$$\varphi_\gamma(x, \theta, z) := \max_{\lambda \in Z} \left\{ f(x, \theta) + \lambda g(x, \theta) - \frac{1}{2\gamma_2} \|\lambda - z\|^2 \right\}.$$

Noticed that $\varphi_\gamma(x, \theta, z)$ can be rewritten as

$$\varphi_\gamma(x, \theta, z) = - \inf_{\lambda \in Z} \left\{ -f(x, \theta) - \lambda g(x, \theta) + \frac{1}{2\gamma_2} \|\lambda - z\|^2 \right\}.$$

By Assumptions 3.2 and 3.3, $f$ and $g$ are both continuous differentiable on an open set containing $X \times Y$, it can be easily shown that $-f(x, \theta) - \lambda g(x, \theta) + \frac{1}{2\gamma_2} \|\lambda - z\|^2$ satisfies the inf-compactness condition in (Bonnans & Shapiro, 2013, Theorem 4.13) on any point $(\bar{x}, \bar{\theta}, \bar{z}) \in X \times Y \times \mathbb{R}^l$, that is, for any $(\bar{x}, \bar{\theta}, \bar{z}) \in X \times Y \times \mathbb{R}^l$, there exist $c \in \mathbb{R}$, compact set $D$ and neighborhood $W$ of $(\bar{x}, \bar{\theta}, \bar{z})$ such that the level set $\{\lambda \in Z \mid -f(x, \theta) - \lambda g(x, \theta) + \frac{1}{\gamma_2} \|\lambda - z\|^2 \leq c\}$ is nonempty and contained in $D$ for any $(x, \theta, z) \in W$. Because $\arg\max_{\lambda \in Z} \left\{ f(x, \theta) + \lambda g(x, \theta) - \frac{1}{2\gamma_2} \|\lambda - z\|^2 \right\}$ is unique for any $(x, \theta, z) \in X \times Y \times \mathbb{R}^l$, we denote it by $\widehat{\lambda}^*(x, \theta, z)$. Then, by Assumptions 3.2 and 3.3, we can derive from (Bonnans & Shapiro, 2013, Theorem 4.13, Remark 4.14) that $\varphi_\gamma(x, \theta, z)$ is differentiable at any point on $X \times Y \times \mathbb{R}^l$, and for any $(x, \theta, z) \in X \times Y \times \mathbb{R}^l$,

$$\nabla \varphi_\gamma(x, \theta, z) = \left( \nabla f(x, \theta) + \widehat{\lambda}^*(x, \theta, z) \nabla g(x, \theta), -(z - \widehat{\lambda}^*(x, \theta, z))/\gamma_2 \right). \tag{22}$$

By simple calculation, we can obtain that

$$\widehat{\lambda}^*(x, \theta, z) := \arg\max_{\lambda \in Z} \left\{ f(x, \theta) + \lambda g(x, \theta) - \frac{1}{2\gamma_2} \|\lambda - z\|^2 \right\} = \mathrm{Proj}_Z \left( z + \gamma_2 g(x, \theta) \right).$$

Since by Assumptions 3.2 and 3.3 that $f$ and $g$ are both continuous differentiable on an open set containing $X \times Y$, and $\mathrm{Proj}_Z$ is continuous, hence $\nabla \varphi_\gamma(x, \theta, z)$ is continuous on $X \times Y \times \mathbb{R}^l$.

Secondly, with the introduced auxiliary function $\varphi_\gamma$, we can rewrite $v_{\gamma,r}$ as

$$v_{\gamma,r}(x, y, z) = \min_{\theta \in Y} \left\{ \varphi_\gamma(x, \theta, z) + \frac{1}{2\gamma_1} \|\theta - y\|^2 \right\}. \tag{23}$$

Next we will show that $\varphi_\gamma(x, y, z)$ is $\rho_f$-weakly convex with respect to variables $(x, y)$ on $X \times Y$ for any fixed $z \in \mathbb{R}^l$. By Assumptions 3.2 and 3.3, we have that for any $\lambda \in \mathbb{R}^l_+$,

$$f(x, y) + \lambda g(x, y) - \frac{1}{2\gamma_2} \|\lambda - z\|^2 + \frac{\rho_f}{2} \|(x, y)\|^2,$$

is convex with respect to variables $(x, y)$ on $X \times Y$. Then by (Rockafellar, 1974, Theorem 1), we obtain that

$$\varphi_\gamma(x, y, z) + \frac{\rho_f}{2} \|(x, y)\|^2 = \max_{\lambda \in Z} \left\{ f(x, y) + \lambda g(x, y) - \frac{1}{2\gamma_2} \|\lambda - z\|^2 + \frac{\rho_f}{2} \|(x, y)\|^2 \right\}$$

is convex with respect to variables $(x, y)$ on $X \times Y$ and thus $\varphi_\gamma(x, y, z)$ is $\rho_f$-weakly convex with respect to variables $(x, y)$ on $X \times Y$ for any fixed $z \in \mathbb{R}^l$. Then, by Assumptions 3.2 and 3.3, it can be easily shown that when $\gamma_1 \in (0, 1/\rho_f)$, $\varphi_\gamma(x, \theta, z) + \frac{1}{2\gamma_1} \|\theta - y\|^2$ satisfies the inf-compactness condition on any point $(\bar{x}, \bar{y}, \bar{z}) \in X \times Y \times \mathbb{R}^l$. Next, because when $\gamma_1 \in (0, 1/\rho_f)$, $\varphi_\gamma(x, \theta, z) + \frac{1}{2\gamma_1} \|\theta - y\|^2$ is strongly convex with respect to $\theta$, $\arg\min_{\theta \in Y} \{ \varphi_\gamma(x, \theta, z) + \frac{1}{2\gamma_1} \|\theta - y\|^2 \}$ is unique and it is equal to $\theta^*(x, y, \lambda)$. By using (Bonnans & Shapiro, 2013, Theorem 4.13, Remark 4.14), the continuous differentiablility of $\varphi_\gamma$ established above and equation (22), we can obtain that $v_{\gamma, r}(x, y, z)$ is differentiable at any point on $X \times Y \times \mathbb{R}^l$, and for any $(x, y, z) \in X \times Y \times \mathbb{R}^l$,

$$\nabla v_{\gamma, r}(x, y, z) = \left( \nabla_x f(x, \theta^*) + \widehat{\lambda}^*(x, \theta^*, z) \nabla_x g(x, \theta^*), (y - \theta^*)/\gamma_1, (\widehat{\lambda}^*(x, \theta^*, z) - z)/\gamma_2 \right),$$

where $\theta^*$ denotes $\theta^*(x, y, z)$.

Finally, noticed that under Assumptions 3.2 and 3.3, when $\gamma_1 \in (0, 1/\rho_f)$ and $\gamma_2 > 0$, the function

$$f(x, \theta) + \lambda g(x, \theta) + \frac{1}{2\gamma_1} \|\theta - y\|^2 - \frac{1}{2\gamma_2} \|\lambda - z\|^2,$$

is strongly convex and strongly concave with respect to $\theta$ and $\lambda$, respectively. Therefore, it follows from saddle point theorem,

$$\min_{\theta \in Y} \max_{\lambda \in Z} \left\{ f(x, \theta) + \lambda g(x, \theta) + \frac{1}{2\gamma_1} \|\theta - y\|^2 - \frac{1}{2\gamma_2} \|\lambda - z\|^2 \right\}$$
$$= \max_{\lambda \in Z} \min_{\theta \in Y} \left\{ f(x, \theta) + \lambda g(x, \theta) + \frac{1}{2\gamma_1} \|\theta - y\|^2 - \frac{1}{2\gamma_2} \|\lambda - z\|^2 \right\},$$

leading to

$$\widehat{\lambda}^*(x, \theta^*, z) = \lambda^*(x, y, z),$$

and thus the conclusion follows. $\qquad\square$

**Remark A.2.** *Using the a similar argument as above, the following result holds for the Lagrangian based proximal value function $v_\gamma(x, y, z)$ when $\gamma_1 \in (0, 1/\rho_f)$ and $\gamma_2 > 0$. That is,*

*(1) The function $v_\gamma(x, y, z)$ is continuously differentiable on $X \times Y \times \mathbb{R}^l$;*

*(2) The gradient of $v_\gamma(x, y, z)$ has closed-form given by*

$$\nabla v_{\gamma, r}(x, y, z) = \left( \nabla_x f(x, \theta^*) + \lambda^* \nabla_x g(x, \theta^*), \frac{(y - \theta^*)}{\gamma_1}, \frac{(\lambda^* - z)}{\gamma_2} \right), \tag{24}$$

*where $\theta^* := \theta^*(x, y, z)$ and $\lambda^* := \lambda^*(x, y, z)$ is the unique saddle point of the following min-max problem:*

$$\min_{\theta \in Y} \max_{\lambda \geq 0} \left\{ f(x, \theta) + \lambda g(x, \theta) + \frac{1}{2\gamma_1} \|\theta - y\|^2 - \frac{1}{2\gamma_2} \|\lambda - z\|^2 \right\}. \tag{25}$$

*(3) Furthermore, for any $\rho_v \geq \rho_f/(1 - \gamma_1 \rho_f)$, $v_\gamma(x, y, z)$ is $\rho_v$-weakly convex with respect to variables $(x, y)$ on $X \times Y$ for any fixed $z \in \mathbb{R}^l$.*

**Lemma A.2.** *Under the assumption of Lemma A.1, let $\gamma_1 \in (0, 1/\rho_f)$, $\gamma_2 > 0$, and $(\bar{x}, \bar{y}, \bar{z}) \in X \times Y \times \mathbb{R}^l$. Then for any $\rho_v \geq \rho_f/(1 - \gamma_1 \rho_f)$ and $(x, y)$ in $X \times Y$, the following inequality holds:*

$$-v_{\gamma, r}(x, y, \bar{z}) \leq -v_{\gamma, r}(\bar{x}, \bar{y}, \bar{z}) - \langle \nabla_{xy} v_{\gamma, r}(\bar{x}, \bar{y}, \bar{z}), (x, y) - (\bar{x}, \bar{y}) \rangle + \frac{\rho_v}{2} \|(x, y) - (\bar{x}, \bar{y})\|^2.$$

*Proof.* The conclusion follows directly from Lemma A.1 that $v_{\gamma,r}(x, y, z)$ is $\rho_v$-weakly convex with respect to variables $(x, y)$ on $X \times Y$ for any fixed $z \in \mathbb{R}^l$. $\qquad\square$

**Lemma A.3.** *Under Assumptions 3.2 and 3.3, let $\gamma_1 \in (0, 1/\rho_f)$, $\gamma_2 > 0$. Then, for any $(x, y, z)$, $(x', y', z') \in X \times \mathbb{R}^m \times \mathbb{R}^l$, the following Lipschitz property holds:*

$$
\begin{aligned}
&\|(\theta^*(x, y, z), \lambda^*(x, y, z)) - (\theta^*(x', y', z'), \lambda^*(x', y', z'))\| \\
&\leq \frac{L_f + C_Z L_{g_2} + L_g}{\rho_T} \|x - x'\| + \frac{1}{\gamma_1 \rho_T} \|y - y'\| + \frac{1}{\gamma_2 \rho_T} \|z - z'\| \\
&\leq L_{\theta\lambda} \|(x, y, z) - (x', y', z')\|,
\end{aligned}
\tag{26}
$$

*where $\rho_T := \min\{1/\gamma_1 - \rho_f, 1/\gamma_2\}$, $C_Z := \max_{z \in Z} \|z\|$ and $L_{\theta\lambda} := \sqrt{3} \max\{L_f + C_Z L_{g_2} + L_g, 1/\gamma_1, 1/\gamma_2\}/\rho_T$.*

*Proof.* For succinctness, we denote $(x, y, z)$ and $(x', y', z')$ by $w$ and $w'$, respectively. Given that $(\theta^*(w), \lambda^*(w))$ is the saddle point for min-max problem

$$
\min_{\theta \in Y} \max_{\lambda \in Z} \left\{ f(x, \theta) + \lambda g(x, \theta) + \frac{1}{2\gamma_1} \|\theta - y\|^2 - \frac{1}{2\gamma_2} \|\lambda - z\|^2 \right\},
$$

it follows from the stationary condition that

$$
\begin{aligned}
0 &\in \nabla_y f(x, \theta^*(w)) + \lambda^*(w) \nabla_y g(x, \theta^*(w)) + (\theta^*(w) - y)/\gamma_1 + \mathcal{N}_Y(\theta^*(w)), \\
0 &\in -g(x, \theta^*(w))) + (\lambda^*(w) - z)/\gamma_2 + \mathcal{N}_Z(\lambda^*(w)).
\end{aligned}
\tag{27}
$$

Under Assumptions 3.2 and 3.3, and that $\gamma_1 \in (0, 1/\rho_f)$ and $\gamma_2 > 0$, we know that the function

$$
f(x, \theta) + \lambda g(x, \theta) + \frac{1}{2\gamma_1} \|\theta - y\|^2 - \frac{1}{2\gamma_2} \|\lambda - z\|^2
$$

is $(1/\gamma_1 - \rho_f)$-strongly convex and $1/\gamma_2$-strongly concave with respect to $\theta$ and $\lambda$, respectively. Then, it follows from (Rockafellar & Wets, 2009, Theorem 12.17 and Exercise 12.59) that the operator

$$
T_w(\theta, \lambda) := (\nabla_y f(x, \theta) + \lambda \nabla_y g(x, \theta) + (\theta - y)/\gamma_1 + \mathcal{N}_Y(\theta), -g(x, \theta) + (\lambda - z)/\gamma_2 + \mathcal{N}_Z(\lambda))
$$

is $\rho_T := \min\{1/\gamma_1 - \rho_f, 1/\gamma_2\}$-strongly monotone. Using $T_w(\theta, \lambda)$, the inclusion (27) can be rewritten as

$$
0 \in T_w(\theta^*(w), \lambda^*(w)).
$$

Similarly, since $(\theta^*(w'), \lambda^*(w'))$ is a saddle point for min-max problem

$$
\min_{\theta \in Y} \max_{\lambda \in Z} \left\{ f(x', \theta) + \lambda g(x', \theta) + \frac{1}{2\gamma_1} \|\theta - y'\|^2 - \frac{1}{2\gamma_2} \|\lambda - z'\|^2 \right\},
$$

we have

$$
0 \in T_{w'}(\theta^*(w'), \lambda^*(w')).
$$

Next, by the definition of $T_w(\theta, \lambda)$, we have

$$
(e_1, e_2) \in T_w(\theta^*(w'), \lambda^*(w')),
$$

with

$$
\begin{aligned}
e_1 &:= \nabla_y f(x, \theta^*(w')) - \nabla_y f(x', \theta^*(w')) \\
&\quad + \lambda^*(w') \left( \nabla_y g(x, \theta^*(w')) - \nabla_y g(x', \theta^*(w')) \right) + (y' - y)/\gamma_1, \\
e_2 &:= -g(x, \theta^*(w'))) + g(x', \theta^*(w'))) + (z' - z)/\gamma_2.
\end{aligned}
$$

Given that $T_w(\theta, \lambda)$ is $\rho_T$-strongly monotone, we have

$$
\begin{aligned}
&\langle -(e_1, e_2), (\theta^*(w), \lambda^*(w)) - (\theta^*(w'), \lambda^*(w')) \rangle \\
&\geq \rho_T \|(\theta^*(w), \lambda^*(w)) - (\theta^*(w'), \lambda^*(w'))\|^2.
\end{aligned}
\tag{28}
$$

For $\lambda^*(w)$, we can obtain that

$$
\begin{aligned}
\lambda^*(w) &= \operatorname*{argmax}_{\lambda \in Z} \left\{ f(x, \theta^*(w)) + \lambda g(x, \theta^*(w)) + \frac{1}{2\gamma_1} \|\theta^*(w) - y\|^2 - \frac{1}{2\gamma_2} \|\lambda - z\|^2 \right\} \\
&= \operatorname{Proj}_Z (z + \gamma_2 g(x, \theta^*(w))).
\end{aligned}
$$

Since $Z$ is a bounded set, it follows that $\lambda^*(w) \leq C_Z$, for any $x \in X, y \in Y, z \in Z$. According to Assumptions 3.2 and 3.3 and the fact that $\theta^*(w) \in Y$, we can derive that

$$\|e_1\| \leq L_f \|x - x'\| + C_Z L_{g_2} \|x - x'\| + \frac{1}{\gamma_1} \|y - y'\|,$$

and

$$\|e_2\| \leq L_g \|x - x'\| + \frac{1}{\gamma_2} \|z - z'\|.$$

Thus, it follows from inequality (28) that

$$\|(\theta^*(w), \lambda^*(w)) - (\theta^*(w'), \lambda^*(w'))\|$$
$$\leq \frac{1}{\rho_T} \|(e_1, e_2)\| \leq \frac{1}{\rho_T} (\|e_1\| + \|e_2\|)$$
$$\leq \frac{L_f + C_Z L_{g_2} + L_g}{\rho_T} \|x - x'\| + \frac{1}{\gamma_1 \rho_T} \|y - y'\| + \frac{1}{\gamma_2 \rho_T} \|z - z'\|,$$

which implies the desired result. $\qquad\square$

**Lemma A.4.** *Suppose Assumptions of Lemma A.3 holds, and let $\gamma_1 \in (0, 1/\rho_f)$, $\gamma_2 > 0$ and $(\bar{x}, \bar{y}, \bar{z}) \in X \times \mathbb{R}^m \times \mathbb{R}^l$. Then for any $z \in \mathbb{R}^l$, we have*

$$-v_{\gamma,r}(\bar{x}, \bar{y}, z) \leq -v_{\gamma,r}(\bar{x}, \bar{y}, \bar{z}) - \langle \nabla_z v_{\gamma,r}(\bar{x}, \bar{y}, \bar{z}), z - \bar{z} \rangle + \frac{L_{v_z}}{2} \|z - \bar{z}\|^2,$$

*with $L_{v_z} := (\gamma_2 \rho_T + 1)/(\gamma_2^2 \rho_T)$.*

*Proof.* By using Lemma A.3, with fixed $(\bar{x}, \bar{y}, \bar{z}) \in X \times \mathbb{R}^m \times \mathbb{R}^l$, for any $z \in \mathbb{R}^l$, we have

$$\|(\theta^*(\bar{x}, \bar{y}, z), \lambda^*(\bar{x}, \bar{y}, z)) - (\theta^*(\bar{x}, \bar{y}, \bar{z}), \lambda^*(\bar{x}, \bar{y}, \bar{z}))\| \leq \frac{1}{\gamma_2 \rho_T} \|z - \bar{z}\|,$$

and thus it follows from Lemma A.1 that

$$\|\nabla_z v_{\gamma,r}(\bar{x}, \bar{y}, z) - \nabla_z v_{\gamma,r}(\bar{x}, \bar{y}, \bar{z})\| = \frac{1}{\gamma_2} \|\lambda^*(\bar{x}, \bar{y}, z) - z - \lambda^*(\bar{x}, \bar{y}, \bar{z}) + \bar{z}\|$$
$$\leq \frac{\gamma_2 \rho_T + 1}{\gamma_2^2 \rho_T} \|z - \bar{z}\|.$$

Then the conclusion follows from (Beck, 2017, Lemma 5.7). $\qquad\square$

**Lemma A.5.** *Under Assumption of Lemma A.3, let $\gamma_1 \in (0, 1/\rho_f)$, $\gamma_2 > 0$ and $\eta_k \in (0, \rho_T/L_B^2)$ with $\rho_T := \min\{1/\gamma_1 - \rho_f, 1/\gamma_2\}$ and $L_B := \max\{L_f + L_g + C_Z L_{g_2} + 1/\gamma_1, L_g + 1/\gamma_2\}$, then, the sequence of $(x^k, y^k, z^k, \theta^k, \lambda^k)$ generated by Algorithm 1 satisfies*

$$\left\|(\theta^{k+1}, \lambda^{k+1}) - (\theta^*(x^k, y^k, z^k), \lambda^*(x^k, y^k, z^k))\right\|$$
$$\leq (1 - \eta_k \rho_T) \left\|(\theta^k, \lambda^k) - (\theta^*(x^k, y^k, z^k), \lambda^*(x^k, y^k, z^k))\right\|. \tag{29}$$

*Proof.* With given $(x^k, y^k, z^k) \in X \times Y \times Z$, we denote $\theta^*(x^k, y^k, z^k)$ and $\lambda^*(x^k, y^k, z^k)$ by $\theta^*$ and $\lambda^*$, respectively, for conciseness. By Assumptions 3.2 and 3.3, and that $\gamma_1 \in (0, 1/\rho_f)$, $\gamma_2 > 0$, we know that

$$f(x^k, \theta) + \lambda g(x^k, \theta) + \frac{1}{2\gamma_1} \|\theta - y^k\|^2 - \frac{1}{2\gamma_2} \|\lambda - z^k\|^2$$

is $(1/\gamma_1 - \rho_f)$-strongly convex and $1/\gamma_2$-strongly concave with respect to $\theta$ and $\lambda$, respectively. Then, the proximal min-max problem in equation (3) is equivalent to finding $(\theta, \lambda)$ satisfying

$$0 \in (A + B)(\theta, \lambda),$$

with

$$A(\theta, \lambda) := \mathcal{N}_Y(\theta) \times \mathcal{N}_Z(\lambda),$$

and

$$B(\theta, \lambda) := \left(\nabla_y f(x^k, \theta) + \lambda \nabla_y g(x^k, \theta) + (\theta - y^k)/\gamma_1, -g(x^k, \theta) + (\lambda - z^k)/\gamma_2\right).$$

And, therefore, $0 \in (A + B)(\theta^*, \lambda^*)$. Because $f(x^k, \theta) + \lambda g(x^k, \theta) + \frac{1}{2\gamma_1}\|\theta - y^k\|^2 - \frac{1}{2\gamma_2}\|\lambda - z^k\|^2$ is $(1/\gamma_1 - \rho_f)$-strongly convex and $1/\gamma_2$-strongly concave with respect to $\theta$ and $\lambda$, respectively, it follows from (Rockafellar & Wets, 2009, Theorem 12.17 and Exercise 12.59) that the operator $B$ is $\rho_T := \min\{1/\gamma_1 - \rho_f, 1/\gamma_2\}$-strongly monotone on $Y \times Z$. And by Assumptions 3.2 and 3.3, we have that for any $(\theta, \lambda), (\theta', \lambda') \in Y \times Z$,

$$
\begin{aligned}
\|B(\theta, \lambda) - B(\theta', \lambda')\| &\leq \|\nabla_y f(x^k, \theta) - \nabla_y f(x^k, \theta')\| + \|\lambda \nabla_y g(x^k, \theta) - \lambda' \nabla_y g(x^k, \theta')\| \\
&\quad + \frac{1}{\gamma_1}\|\theta - \theta'\| + \|g(x^k, \theta) - g(x^k, \theta')\| + \frac{1}{\gamma_2}\|\lambda - \lambda'\| \\
&\leq \left(L_f + L_g + \frac{1}{\gamma_1}\right)\|\theta - \theta'\| + \frac{1}{\gamma_2}\|\lambda - \lambda'\| \\
&\quad + \|\lambda \nabla_y g(x^k, \theta) - \lambda' \nabla_y g(x^k, \theta)\| + \|\lambda' \nabla_y g(x^k, \theta) - \lambda' \nabla_y g(x^k, \theta')\| \\
&\leq \left(L_f + L_g + C_Z L_{g_2} + \frac{1}{\gamma_1}\right)\|\theta - \theta'\| + \left(L_g + \frac{1}{\gamma_2}\right)\|\lambda - \lambda'\|,
\end{aligned}
$$

where the last inequality follows from the fact that $C_Z := \max_{z \in Z}\|z\|$ and $\max_{x \in X, \theta \in Y}\|\nabla_y g(x, \theta)\| \leq L_g$. Therefore, we obtain that the operator $B$ is $L_B := \max\{L_f + L_g + C_Z L_{g_2} + 1/\gamma_1, L_g + 1/\gamma_2\}$-Lipschitz continuous. Then, we can have from (Bauschke & Combettes, 2011, Proposition 26.1(iv)) that $(\theta^*, \lambda^*) = (Id + \eta_k A)^{-1}((\theta^*, \lambda^*) - \eta_k B(\theta^*, \lambda^*))$, with $Id$ denoting the identity operator. Since $(\theta^{k+1}, \lambda^{k+1}) = (Id + \eta_k A)^{-1}((\theta^k, \lambda^k) - \eta_k B(\theta^k, \lambda^k))$, as shown in the proof of (Bauschke & Combettes, 2011, Proposition 26.9) that when $\eta_k \in (0, 2\rho_T/L_B^2)$,

$$\|(\theta^{k+1}, \lambda^{k+1}) - (\theta^*, \lambda^*)\| \leq \left(1 - \eta_k(2\rho_T - \eta_k L_B^2)\right)\|(\theta^k, \lambda^k) - (\theta^*, \lambda^*)\|.$$

When $\eta_k \in (0, \rho_T/L_B^2)$, we can obtain from the above inequality that

$$\|(\theta^{k+1}, \lambda^{k+1}) - (\theta^*, \lambda^*)\| \leq (1 - \eta_k \rho_T)\|(\theta^k, \lambda^k) - (\theta^*, \lambda^*)\|.$$

$\square$

As previously stated that the update of variables $(x, y, z)$ in equation (10) can be interpreted as inexact alternating proximal gradient from $(x^k, y^k, z^k)$ on $\min_{(x,y) \in C, z \in Z} \phi_{c_k}(x, y, z)$, in which $\phi_{c_k}$ is defined in equation (13) as

$$\phi_{c_k}(x, y, z) := \frac{1}{c_k}\left(F(x, y) - \underline{F}\right) + f(x, y) - v_{\gamma, r}(x, y, z).$$

Since $v_{\gamma, r}(x, y, z) \leq v_\gamma(x, y, z) \leq f(x, y)$ for all $(x, y) \in C$, by Assumption 3.1, we have $\phi_{c_k}(x, y, z) \geq 0$ for all $(x, y, z) \in C \times \mathbb{R}^l$. In the following lemma, we demonstrate that the function $\phi_{c_k}(x, y, z)$ exhibits a decreasing property with errors at each iteration.

**Lemma A.6.** *Under Assumptions 3.1, 3.2 and 3.3, let $\gamma_1 \in (0, 1/\rho_f)$ and $\gamma_2 > 0$. Then the sequence of $(x^k, y^k, \theta^k)$ generated by Algorithm 1 satisfies*

$$
\begin{aligned}
\phi_{c_k}(x^{k+1}, y^{k+1}, z^{k+1}) \leq{}& \phi_{c_k}(x^k, y^k, z^k) \\
&- \left(\frac{1}{2\alpha_k} - \frac{L_{\phi_k}}{2} - \frac{\beta_k L_{\theta\lambda}^2}{\gamma_2^2}\right)\|(x^{k+1}, y^{k+1}) - (x^k, y^k)\|^2 \\
&- \left(\frac{1}{2\beta_k} - \frac{L_{v_z}}{2}\right)\|z^{k+1} - z^k\|^2 \\
&+ \left(\alpha_k L_g^2 + \frac{\beta_k}{\gamma_2^2}\right)\left\|\lambda^{k+1} - \lambda^*(x^k, y^k, z^k)\right\|^2 \\
&+ \frac{\alpha_k}{2}\left(2(L_f + C_Z L_{g_1})^2 + \frac{1}{\gamma_1^2}\right)\left\|\theta^{k+1} - \theta^*(x^k, y^k, z^k)\right\|^2,
\end{aligned}
$$

(30)

*where $L_{\phi_k} := L_F/c_k + L_f + \rho_v$.*

*Proof.* Given Assumptions 3.1 and 3.2 that $\nabla F$ and $\nabla f$ are $L_F$- and $L_f$-Lipschitz continuous on $X \times Y$, respectively, and applying (Beck, 2017, Lemma 5.7) and Lemma A.2, we obtain

$$
\phi_{c_k}(x^{k+1}, y^{k+1}, z^k) \le \phi_{c_k}(x^k, y^k, z^k) + \langle \nabla_{xy}\phi_{c_k}(x^k, y^k, z^k), (x^{k+1}, y^{k+1}) - (x^k, y^k) \rangle \\
+ \frac{L_{\phi_k}}{2} \|(x^{k+1}, y^{k+1}) - (x^k, y^k)\|^2, \tag{31}
$$

with $L_{\phi_k} := L_F/c_k + L_f + \rho_v$. Based on the update rule of variable $(x, y)$ in equation (10), the convexity of $C := \{(x, y) \in X \times Y \mid g(x, y) \le 0\}$ and the property of the projection operator $\mathrm{Proj}_C$, we have

$$
\langle (x^k, y^k) - \alpha_k(d_x^k, d_y^k) - (x^{k+1}, y^{k+1}), (x^k, y^k) - (x^{k+1}, y^{k+1}) \rangle \le 0,
$$

leading to

$$
\langle (d_x^k, d_y^k), (x^{k+1}, y^{k+1}) - (x^k, y^k) \rangle \le -\frac{1}{\alpha_k} \|(x^{k+1}, y^{k+1}) - (x^k, y^k)\|^2.
$$

Combining this with inequality (31), we infer that

$$
\phi_{c_k}(x^{k+1}, y^{k+1}, z^k) \le \phi_{c_k}(x^k, y^k, z^k) - \left( \frac{1}{\alpha_k} - \frac{L_{\phi_k}}{2} \right) \|(x^{k+1}, y^{k+1}) - (x^k, y^k)\|^2 \\
+ \langle \nabla_{xy}\phi_{c_k}(x^k, y^k, z^k) - (d_x^k, d_y^k), (x^{k+1}, y^{k+1}) - (x^k, y^k) \rangle. \tag{32}
$$

It should be noticed that since $(x^k, y^k, z^k) \in X \times Y \times Z$ and $(\theta^*(x^k, y^k, z^k), \lambda^*(x^k, y^k, z^k))$, $(\theta^k, \lambda^k) \in Y \times Z$ for all $k$, it holds that $\lambda^k \in Z$ and $\|\nabla_x g(x^k, \theta^*(x^k, y^k, z^k))\| \le \max_{x \in X, \theta \in Y} \|\nabla_x g(x, \theta)\| \le L_g$ for all $k$. Considering the formula of $\nabla_{xy} v_{\gamma, r}(x, y, z)$ derived in Lemma A.2 and the definitions of $d_x^k$ and $d_y^k$ provided in equation (11), we can obtain that

$$
\left\| \nabla_{xy}\phi_{c_k}(x^k, y^k, z^k) - (d_x^k, d_y^k) \right\|^2
$$
$$
= \left\| \nabla_x f(x^k, \theta^*(x^k, y^k, z^k)) + \lambda^*(x^k, y^k, z^k)\nabla_x g(x^k, \theta^*(x^k, y^k, z^k)) \right.
$$
$$
\left. - \nabla_x f(x^k, \theta^{k+1}) - \lambda^{k+1}\nabla_x g(x^k, \theta^{k+1}) \right\|^2 + \frac{1}{\gamma_1^2} \|\theta^*(x^k, y^k, z^k) - \theta^{k+1}\|^2
$$
$$
\le 2 \left\| \nabla_x f(x^k, \theta^*(x^k, y^k, z^k)) + \lambda^{k+1}\nabla_x g(x^k, \theta^*(x^k, y^k, z^k)) \right.
$$
$$
\left. - \nabla_x f(x^k, \theta^{k+1}) - \lambda^{k+1}\nabla_x g(x^k, \theta^{k+1}) \right\|^2 \tag{33}
$$
$$
+ 2\|\lambda^*(x^k, y^k, z^k)\nabla_x g(x^k, \theta^*(x^k, y^k, z^k)) - \lambda^{k+1}\nabla_x g(x^k, \theta^*(x^k, y^k, z^k))\|^2
$$
$$
+ \frac{1}{\gamma_1^2} \|\theta^*(x^k, y^k, z^k) - \theta^{k+1}\|^2
$$
$$
\le \left( 2(L_f + C_Z L_{g_1})^2 + \frac{1}{\gamma_1^2} \right) \left\| \theta^{k+1} - \theta^*(x^k, y^k, z^k) \right\|^2 + 2L_g^2 \left\| \lambda^{k+1} - \lambda^*(x^k, y^k, z^k) \right\|^2,
$$

where the last inequality follows from Assumptions 3.2 and 3.3, $\|\lambda^{k+1}\| \le C_Z$ and $\|\nabla_x g(x^k, \theta^*(x^k, y^k, z^k))\| \le L_g$. This yields

$$
\langle \nabla_{xy}\phi_{c_k}(x^k, y^k, z^k) - (d_x^k, d_y^k), (x^{k+1}, y^{k+1}) - (x^k, y^k) \rangle
$$
$$
\le \frac{\alpha_k}{2} \left( 2(L_f + C_Z L_{g_1})^2 + 1/\gamma_1^2 \right) \left\| \theta^{k+1} - \theta^*(x^k, y^k, z^k) \right\|^2 + \alpha_k L_g^2 \left\| \lambda^{k+1} - \lambda^*(x^k, y^k, z^k) \right\|^2
$$
$$
+ \frac{1}{2\alpha_k} \|(x^{k+1}, y^{k+1}) - (x^k, y^k)\|^2,
$$

which combing with inequality (32) leads to

$$
\phi_{c_k}(x^{k+1}, y^{k+1}, z^k)
$$
$$
\le \phi_{c_k}(x^k, y^k, z^k) - \left( \frac{1}{2\alpha_k} - \frac{L_{\phi_k}}{2} \right) \|(x^{k+1}, y^{k+1}) - (x^k, y^k)\|^2
$$
$$
+ \frac{\alpha_k}{2} \left( 2(L_f + C_Z L_{g_1})^2 + \frac{1}{\gamma_1^2} \right) \left\| \theta^{k+1} - \theta^*(x^k, y^k, z^k) \right\|^2 \tag{34}
$$
$$
+ \alpha_k L_g^2 \left\| \lambda^{k+1} - \lambda^*(x^k, y^k, z^k) \right\|^2.
$$

According to the update rule of variable $z$ in equation (10) and the property of the projection operator $\text{Proj}_Z$, we have

$$\langle d_z^k, z^{k+1} - z^k \rangle \leq -\frac{1}{\beta_k} \|z^{k+1} - z^k\|^2. \tag{35}$$

Using Lemma A.4, we obtain

$$
\begin{aligned}
&\phi_{c_k}(x^{k+1}, y^{k+1}, z^{k+1}) \\
&\leq \phi_{c_k}(x^{k+1}, y^{k+1}, z^k) + \langle \nabla_z \phi_{c_k}(x^{k+1}, y^{k+1}, z^k), z^{k+1} - z^k \rangle + \frac{L_{v_z}}{2} \|z^{k+1} - z^k\|^2.
\end{aligned}
\tag{36}
$$

Combining this with inequality (35), we can derive

$$
\begin{aligned}
&\phi_{c_k}(x^{k+1}, y^{k+1}, z^{k+1}) \\
&\leq \phi_{c_k}(x^{k+1}, y^{k+1}, z^k) - \left( \frac{1}{\beta_k} - \frac{L_{v_z}}{2} \right) \|z^{k+1} - z^k\|^2 \\
&\quad + \langle \nabla_z \phi_{c_k}(x^{k+1}, y^{k+1}, z^k) - d_z^k, z^{k+1} - z^k \rangle.
\end{aligned}
\tag{37}
$$

Using the definition of $\phi_{c_k}$ and the formula of $\nabla_z v_{\gamma,r}$ derived in Lemma A.1 and the definition of $d_z^k$ provided in equation (11), we have

$$
\begin{aligned}
\left\| \nabla_z \phi_{c_k}(x^{k+1}, y^{k+1}, z^k) - d_z^k \right\|^2 &= \left\| -\nabla_z v_{\gamma,r}(x^{k+1}, y^{k+1}, z^k) - d_z^k \right\|^2 \\
&= \left\| (z^k - \lambda^*(x^{k+1}, y^{k+1}, z^k))/\gamma_2 - (z^k - \lambda^{k+1})/\gamma_2 \right\|^2 \\
&= \frac{1}{\gamma_2^2} \left\| \lambda^{k+1} - \lambda^*(x^{k+1}, y^{k+1}, z^k) \right\|^2,
\end{aligned}
$$

and thus

$$
\begin{aligned}
\langle \nabla_z \phi_{c_k}(x^{k+1}, y^{k+1}, z^k) - d_z^k, z^{k+1} - z^k \rangle &\leq \frac{\beta_k}{2\gamma_2^2} \left\| \lambda^{k+1} - \lambda^*(x^{k+1}, y^{k+1}, z^k) \right\|^2 \\
&\quad + \frac{1}{2\beta_k} \|z^{k+1} - z^k\|^2.
\end{aligned}
$$

Then, we have from inequality (37) that

$$
\begin{aligned}
&\phi_{c_k}(x^{k+1}, y^{k+1}, z^{k+1}) \\
&\leq \phi_{c_k}(x^{k+1}, y^{k+1}, z^k) - \left( \frac{1}{2\beta_k} - \frac{L_{v_z}}{2} \right) \|z^{k+1} - z^k\|^2 \\
&\quad + \frac{\beta_k}{2\gamma_2^2} \left\| \lambda^{k+1} - \lambda^*(x^{k+1}, y^{k+1}, z^k) \right\|^2 \\
&\leq \phi_{c_k}(x^{k+1}, y^{k+1}, z^k) - \left( \frac{1}{2\beta_k} - \frac{L_{v_z}}{2} \right) \|z^{k+1} - z^k\|^2 + \frac{\beta_k}{\gamma_2^2} \left\| \lambda^{k+1} - \lambda^*(x^k, y^k, z^k) \right\|^2 \\
&\quad + \frac{\beta_k L_{\theta\lambda}^2}{\gamma_2^2} \|(x^{k+1}, y^{k+1}) - (x^k, y^k)\|^2.
\end{aligned}
\tag{38}
$$

where the last inequality follows from Lemma A.3. The conclusion follows by combining estimates (34) and (38). $\qquad\square$

## A.5 Proof of Lemma 3.1

By utilizing the auxiliary lemmas established in the previous section, we will demonstrate the decreasing property of

$$V_k := \phi_{c_k}(x^k, y^k, z^k) + C_{\theta\lambda} \left\| (\theta^k, \lambda^k) - (\theta^*(x^k, y^k, z^k), \lambda^*(x^k, y^k, z^k)) \right\|^2, \tag{39}$$

where $C_{\theta\lambda} := \max\{(L_f + C_Z L_{g_1})^2 + 1/(2\gamma_1^2) + L_g^2, 1/\gamma_2^2\}$, and

$$\phi_{c_k}(x, y, z) := \frac{1}{c_k} \big( F(x, y) - \underline{F} \big) + f(x, y) - v_{\gamma,r}(x, y, z). \tag{13}$$

**Lemma A.7.** *Under Assumptions 3.1, 3.2 and 3.3, let $\gamma_1 \in (0, 1/\rho_f)$, $\gamma_2 > 0$, $c_{k+1} \geq c_k$ and $\eta_k \in (\underline{\eta}, \rho_T/L_B^2)$ with $\underline{\eta} > 0$, $\rho_T := \min\{1/\gamma_1 - \rho_f, 1/\gamma_2\}$ and $L_B := \max\{L_f + L_g + C_Z L_{g_2} + 1/\gamma_1, L_g + 1/\gamma_2\}$, then there exist constants $c_\alpha, c_\beta > 0$ such that when $0 < \alpha_k \leq c_\alpha$ and $0 < \beta_k \leq c_\beta$, the sequence of $(x^k, y^k, z^k)$ generated by Algorithm 1 satisfies*

$$
\begin{aligned}
V_{k+1} - V_k \leq &-\frac{1}{4\alpha_k} \|(x^{k+1}, y^{k+1}) - (x^k, y^k)\|^2 - \frac{1}{4\beta_k} \|z^{k+1} - z^k\|^2 \\
&- \underline{\eta}\rho_T C_{\theta\lambda} \left\| (\theta^k, \lambda^k) - (\theta^*(x^k, y^k, z^k), \lambda^*(x^k, y^k, z^k)) \right\|^2.
\end{aligned}
\tag{40}
$$

*Proof.* For succinctness, we denote $(x^k, y^k, z^k)$ by $w^k$. Let us first recall estimate (30) from Lemma A.6, which states that

$$
\begin{aligned}
\phi_{c_k}(w^{k+1}) \leq &\phi_{c_k}(w^k) - \left( \frac{1}{2\alpha_k} - \frac{L_{\phi_k}}{2} - \frac{\beta_k L_{\theta\lambda}^2}{\gamma_2^2} \right) \|(x^{k+1}, y^{k+1}) - (x^k, y^k)\|^2 \\
&- \left( \frac{1}{2\beta_k} - \frac{L_{v_z}}{2} \right) \|z^{k+1} - z^k\|^2 + \left( \alpha_k L_g^2 + \frac{\beta_k}{\gamma_2^2} \right) \left\| \lambda^{k+1} - \lambda^*(w^k) \right\|^2 \\
&+ \frac{\alpha_k}{2} \left( 2(L_f + C_Z L_{g_1})^2 + \frac{1}{\gamma_1^2} \right) \left\| \theta^{k+1} - \theta^*(w^k) \right\|^2.
\end{aligned}
\tag{41}
$$

Since $c_{k+1} \geq c_k$, we can infer that $(F(x^{k+1}, y^{k+1}) - \underline{F})/c_{k+1} \leq (F(x^{k+1}, y^{k+1}) - \underline{F})/c_k$. Combining with inequality (41) leads to

$$
\begin{aligned}
V_{k+1} - V_k = &\phi_{c_{k+1}}(w^{k+1}) - \phi_{c_k}(w^k) \\
&+ C_{\theta\lambda} \left\| (\theta^{k+1}, \lambda^{k+1}) - (\theta^*(w^{k+1}), \lambda^*(w^{k+1})) \right\|^2 - C_{\theta\lambda} \left\| (\theta^k, \lambda^k) - (\theta^*(w^k), \lambda^*(w^k)) \right\|^2 \\
\leq &\phi_{c_k}(x^{k+1}, y^{k+1}, z^{k+1}) - \phi_{c_k}(x^k, y^k, z^k) \\
&+ C_{\theta\lambda} \left\| (\theta^{k+1}, \lambda^{k+1}) - (\theta^*(w^{k+1}), \lambda^*(w^{k+1})) \right\|^2 - C_{\theta\lambda} \left\| (\theta^k, \lambda^k) - (\theta^*(w^k), \lambda^*(w^k)) \right\|^2 \\
\leq &- \left( \frac{1}{2\alpha_k} - \frac{L_{\phi_k}}{2} - \frac{\beta_k L_{\theta\lambda}^2}{\gamma_2^2} \right) \|(x^{k+1}, y^{k+1}) - (x^k, y^k)\|^2 \\
&- \left( \frac{1}{2\beta_k} - \frac{L_{v_z}}{2} \right) \|z^{k+1} - z^k\|^2 + C_{\theta\lambda} \left\| (\theta^{k+1}, \lambda^{k+1}) - (\theta^*(w^{k+1}), \lambda^*(w^{k+1})) \right\|^2 \\
&- C_{\theta\lambda} \left\| (\theta^k, \lambda^k) - (\theta^*(w^k), \lambda^*(w^k)) \right\|^2 + \left( \alpha_k L_g^2 + \frac{\beta_k}{\gamma_2^2} \right) \left\| \lambda^{k+1} - \lambda^*(w^k) \right\|^2 \\
&+ \frac{\alpha_k}{2} \left( 2(L_f + C_Z L_{g_1})^2 + \frac{1}{\gamma_1^2} \right) \left\| \theta^{k+1} - \theta^*(w^k) \right\|^2 \\
\leq &- \left( \frac{1}{2\alpha_k} - \frac{L_{\phi_k}}{2} - \frac{\beta_k L_{\theta\lambda}^2}{\gamma_2^2} \right) \|(x^{k+1}, y^{k+1}) - (x^k, y^k)\|^2 - \left( \frac{1}{2\beta_k} - \frac{L_{v_z}}{2} \right) \|z^{k+1} - z^k\|^2 \\
&+ C_{\theta\lambda} \left\| (\theta^{k+1}, \lambda^{k+1}) - (\theta^*(w^{k+1}), \lambda^*(w^{k+1})) \right\|^2 - C_{\theta\lambda} \left\| (\theta^k, \lambda^k) - (\theta^*(w^k), \lambda^*(w^k)) \right\|^2 \\
&+ 2\max\{\alpha_k, \beta_k\} C_{\theta\lambda} \left\| (\theta^{k+1}, \lambda^{k+1}) - (\theta^*(w^k), \lambda^*(w^k)) \right\|^2,
\end{aligned}
\tag{42}
$$

where the last inequality follows from the fact that

$$
C_{\theta\lambda} := \max \left\{ (L_f + C_Z L_{g_1})^2 + 1/(2\gamma_1^2) + L_g^2, 1/\gamma_2^2 \right\}.
$$

We can demonstrate that

$$
\begin{aligned}
&\left\| (\theta^{k+1}, \lambda^{k+1}) - (\theta^*(w^{k+1}), \lambda^*(w^{k+1})) \right\|^2 - \left\| (\theta^k, \lambda^k) - (\theta^*(w^k), \lambda^*(w^k)) \right\|^2 \\
&+ 2\alpha_k \left\| (\theta^{k+1}, \lambda^{k+1}) - (\theta^*(w^k), \lambda^*(w^k)) \right\|^2 \\
\leq &(1 + \epsilon_k + 2\alpha_k) \left\| (\theta^{k+1}, \lambda^{k+1}) - (\theta^*(w^k), \lambda^*(w^k)) \right\|^2 - \left\| (\theta^k, \lambda^k) - (\theta^*(w^k), \lambda^*(w^k)) \right\|^2 \\
&+ (1 + \frac{1}{\epsilon_k}) \|(\theta^*(w^{k+1}), \lambda^*(w^{k+1})) - (\theta^*(w^k), \lambda^*(w^k))\|^2 \\
\leq &(1 + \epsilon_k + 2\alpha_k)(1 - \eta_k \rho_T)^2 \|(\theta^k, \lambda^k) - (\theta^*(w^k), \lambda^*(w^k))\|^2 - \left\| (\theta^k, \lambda^k) - (\theta^*(w^k), \lambda^*(w^k)) \right\|^2 \\
&+ (1 + \frac{1}{\epsilon_k}) L_{\theta\lambda}^2 \left\| w^{k+1} - w^k \right\|^2,
\end{aligned}
$$

for any $\epsilon_k > 0$, where the second inequality is a consequence of Lemmas A.3 and A.5. By setting $\epsilon_k = \eta_k \rho_T / 2$ in the above inequality, we obtain that when $\alpha_k \leq \eta_k \rho_T / 4$, it holds that $(1 + \epsilon_k + 2\alpha_k)(1 - \eta_k \rho_T) \leq 1$ and thus

$$
\begin{aligned}
&\left\| (\theta^{k+1}, \lambda^{k+1}) - (\theta^*(w^{k+1}), \lambda^*(w^{k+1})) \right\|^2 - \left\| (\theta^k, \lambda^k) - (\theta^*(w^k), \lambda^*(w^k)) \right\|^2 \\
&+ 2\alpha_k \left\| (\theta^{k+1}, \lambda^{k+1}) - (\theta^*(w^k), \lambda^*(w^k)) \right\|^2 \\
&\leq -\eta_k \rho_T \| (\theta^k, \lambda^k) - (\theta^*(w^k), \lambda^*(w^k)) \|^2 + \left( 1 + \frac{2}{\eta_k \rho_T} \right) L_{\theta\lambda}^2 \left\| w^{k+1} - w^k \right\|^2,
\end{aligned} \tag{43}
$$

Similarly, we can show that when $\beta_k \leq \eta_k \rho_T / 4$, it holds that

$$
\begin{aligned}
&\left\| (\theta^{k+1}, \lambda^{k+1}) - (\theta^*(w^{k+1}), \lambda^*(w^{k+1})) \right\|^2 - \left\| (\theta^k, \lambda^k) - (\theta^*(w^k), \lambda^*(w^k)) \right\|^2 \\
&+ 2\beta_k \left\| (\theta^{k+1}, \lambda^{k+1}) - (\theta^*(w^k), \lambda^*(w^k)) \right\|^2 \\
&\leq -\eta_k \rho_T \| (\theta^k, \lambda^k) - (\theta^*(w^k), \lambda^*(w^k)) \|^2 + \left( 1 + \frac{2}{\eta_k \rho_T} \right) L_{\theta\lambda}^2 \left\| w^{k+1} - w^k \right\|^2.
\end{aligned} \tag{44}
$$

Combining estimates (42), (43) and (44), we have

$$
\begin{aligned}
V_{k+1} - V_k \leq &- \left( \frac{1}{2\alpha_k} - \frac{L_{\phi_k}}{2} - \frac{\beta_k L_{\theta\lambda}^2}{\gamma_2^2} - \left( 1 + \frac{2}{\eta_k \rho_T} \right) L_{\theta\lambda}^2 C_{\theta\lambda} \right) \left\| (x^{k+1}, y^{k+1}) - (x^k, y^k) \right\|^2 \\
&- \left( \frac{1}{2\beta_k} - \frac{L_{v_z}}{2} - \left( 1 + \frac{2}{\eta_k \rho_T} \right) L_{\theta\lambda}^2 C_{\theta\lambda} \right) \| z^{k+1} - z^k \|^2 \\
&- \eta_k \rho_T C_{\theta\lambda} \| (\theta^k, \lambda^k) - (\theta^*(w^k), \lambda^*(w^k)) \|^2.
\end{aligned} \tag{45}
$$

When $c_{k+1} \geq c_k$, $\eta_k \geq \underline{\eta} > 0$, $\alpha_k \leq \underline{\eta} \rho_T / 4$ and $\beta_k \leq \underline{\eta} \rho_T / 4$, it holds that for any $k$, $\alpha_k \leq \eta_k \rho_T / 4$, $\beta_k \leq \eta_k \rho_T / 4$,

$$
\frac{L_{\phi_k}}{2} + \frac{\beta_k L_{\theta\lambda}^2}{\gamma_2^2} + \left( 1 + \frac{2}{\eta_k \rho_T} \right) L_{\theta\lambda}^2 C_{\theta\lambda} \leq \frac{L_{\phi_0}}{2} + + \frac{\underline{\eta} \rho_T L_{\theta\lambda}^2}{4\gamma_2^2} + \left( 1 + \frac{2}{\underline{\eta} \rho_T} \right) L_{\theta\lambda}^2 C_{\theta\lambda} =: C_\alpha, \tag{46}
$$

and

$$
\frac{L_{v_z}}{2} + \left( 1 + \frac{2}{\eta_k \rho_T} \right) L_{\theta\lambda}^2 C_{\theta\lambda} \leq \frac{L_{v_z}}{2} + \left( 1 + \frac{2}{\underline{\eta} \rho_T} \right) L_{\theta\lambda}^2 C_{\theta\lambda} =: C_\beta, \tag{47}
$$

Consequently, if $c_\alpha, c_\beta > 0$ satisfies

$$
c_\alpha \leq \min \left\{ \frac{1}{4} \underline{\eta} \rho_T, \frac{1}{4C_\alpha} \right\}, \qquad c_\beta \leq \min \left\{ \frac{1}{4} \underline{\eta} \rho_T, \frac{1}{4C_\beta} \right\}, \tag{48}
$$

then, when $0 < \alpha_k \leq c_\alpha$ and $0 < \beta_k \leq c_\beta$, it holds that

$$
\frac{1}{2\alpha_k} - \frac{L_{\phi_k}}{2} - \frac{\beta_k L_{\theta\lambda}^2}{\gamma_2^2} - \left( 1 + \frac{2}{\eta_k \rho_T} \right) L_{\theta\lambda}^2 C_{\theta\lambda} \geq \frac{1}{4\alpha_k},
$$

and

$$
\frac{1}{2\beta_k} - \frac{L_{v_z}}{2} - \left( 1 + \frac{2}{\eta_k \rho_T} \right) L_{\theta\lambda}^2 C_{\theta\lambda} \geq \frac{1}{4\beta_k}.
$$

Then, the conclusion follows from estimate (45). $\qquad \square$

## A.6 PROOF OF THEOREM 3.1

We establish the non-asymptotic convergence of LV-HBA in the following theorem, as measured by the residual function defined in equation (15),

$$
R_k := \mathrm{dist}\, (0, (\nabla F(x, y), 0) + c_k ((\nabla f(x, y), 0) - \nabla v_{\gamma, r}(x, y, z)) + \mathcal{N}_{C \times Z}(x, y, z)). \tag{15}
$$

**Theorem A.3.** *Under Assumptions of Lemma A.7, let $\gamma_1 \in (0, 1/\rho_f)$, $\gamma_2 > 0$, $c_k = \underline{c}(k+1)^p$ with $p \in (0, 1/2), \underline{c} > 0$, and $\eta_k \in (\underline{\eta}, \rho_T/L_B^2)$ with $\underline{\eta} > 0$, then there exists $c_\alpha, c_\beta > 0$ such that when $\alpha_k \in (\underline{\alpha}, c_\alpha)$ and $\beta_k \in (\underline{\beta}, c_\beta)$ with $\underline{\alpha}, \underline{\beta} > 0$, the sequence of $(x^k, y^k, z^k, \theta^k, \lambda^k)$ generated by LV-HBA in Algorithm 1 satisfies*

$$\min_{0 \le k \le K} \left\| (\theta^k, \lambda^k) - (\theta_r^*(x^k, y^k, z^k), \lambda_r^*(x^k, y^k, z^k)) \right\| = O\left(\frac{1}{K^{1/2}}\right), \tag{49}$$

*and*

$$\min_{0 \le k \le K} R_k(x^{k+1}, y^{k+1}, z^{k+1}) = O\left(\frac{1}{K^{(1-2p)/2}}\right). \tag{50}$$

*Furthermore, if there exists $M > 0$ such that $\psi_{c_k}(x^k, y^k, z^k) \le M$ for any $k$, the sequence of $(x^k, y^k, z^k)$ also satisfies*

$$0 \le f(x^k, y^k) - v_\gamma(x^k, y^k, z^k) \le f(x^k, y^k) - v_{\gamma,r}(x^k, y^k, z^k) = O\left(\frac{1}{K^p}\right). \tag{51}$$

*Proof.* Firstly, given $c_\alpha, c_\beta > 0$ in equation (48), Lemma 3.1 guarantees that the inequality (14) holds when $\alpha_k \le c_\alpha$, $\beta_k \le c_\beta$. By telescoping the inequality (14) for $k = 0, 1, \ldots, K-1$, we get

$$\sum_{k=0}^{K-1} \left( \frac{1}{4\alpha_k} \| (x^{k+1}, y^{k+1}) - (x^k, y^k) \|^2 + \frac{1}{4\beta_k} \| z^{k+1} - z^k \|^2 \right.$$
$$\left. + \underline{\eta}\rho_T C_{\theta\lambda} \left\| (\theta^k, \lambda^k) - (\theta_r^*(x^k, y^k, z^k), \lambda_r^*(x^k, y^k, z^k)) \right\|^2 \right) \tag{52}$$

$$\le V_0 - V_K \le V_0,$$

where the last inequality is valid since $V_K$ is nonnegative. The latter is implies by the fact that $v_{\gamma,r}(x, y, z) \le v_\gamma(x, y, z) \le f(x, y)$ for all $(x, y) \in C$. Thus, we have

$$\sum_{k=0}^{\infty} \left\| (\theta^k, \lambda^k) - (\theta_r^*(x^k, y^k, z^k), \lambda_r^*(x^k, y^k, z^k)) \right\|^2 < \infty. \tag{53}$$

This implies that the estimate (49) holds, that is,

$$\min_{0 \le k \le K} \left\| (\theta^k, \lambda^k) - (\theta_r^*(x^k, y^k, z^k), \lambda_r^*(x^k, y^k, z^k)) \right\| = O\left(\frac{1}{K^{1/2}}\right). \tag{54}$$

Secondly, According to the update rule of variables $(x, y, z)$ in equation (10), we have that

$$0 \in c_k(d_x^k, d_y^k) + \mathcal{N}_C(x^{k+1}, y^{k+1}) + \frac{c_k}{\alpha_k} \left( (x^{k+1}, y^{k+1}) - (x^k, y^k) \right),$$
$$0 \in c_k d_z^k + \mathcal{N}_Z(z^{k+1}) + \frac{c_k}{\beta_k} \left( z^{k+1} - z^k \right). \tag{55}$$

By the definitions of $d_x^k, d_y^k$ and $d_z^k$ given in equation (11), we obtain

$$(e_{xy}^k, e_z^k) \in \left( \nabla F(x^{k+1}, y^{k+1}), 0 \right) + c_k \left( (\nabla f(x^{k+1}, y^{k+1}), 0) - \nabla v_{\gamma,r}(x^{k+1}, y^{k+1}, z^{k+1}) \right)$$
$$+ \mathcal{N}_{C \times Z}(x^{k+1}, y^{k+1}, z^{k+1}),$$

with

$$e_{xy}^k := \nabla_{xy} \psi_{c_k}(x^{k+1}, y^{k+1}, z^{k+1}) - c_k(d_x^k, d_y^k) - \frac{c_k}{\alpha_k} \left( (x^{k+1}, y^{k+1}) - (x^k, y^k) \right),$$
$$e_z^k := \nabla_z \psi_{c_k}(x^{k+1}, y^{k+1}, z^{k+1}) - c_k d_z^k - \frac{c_k}{\beta_k} \left( z^{k+1} - z^k \right). \tag{56}$$

Next, we estimate $\|e_{xy}^k\|$. We have

$$\|e_{xy}^k\| \le \|\nabla_{xy} \psi_{c_k}(x^{k+1}, y^{k+1}, z^{k+1}) - \nabla_{xy} \psi_{c_k}(x^k, y^k, z^k)\| + \|\nabla_{xy} \psi_{c_k}(x^k, y^k, z^k) - c_k(d_x^k, d_y^k)\|$$
$$+ \frac{c_k}{\alpha_k} \left\| (x^{k+1}, y^{k+1}) - (x^k, y^k) \right\|.$$

For the first term in the right hand side of the above inequality, by using Assumptions 3.2, 3.2 and 3.3, Lemmas A.1 and A.3, we can obtain the existence of $L_{\psi_1} > 0$ such that

$$\|\nabla_{xy}\psi_{c_k}(x^{k+1}, y^{k+1}, z^{k+1}) - \nabla_{xy}\psi_{c_k}(x^k, y^k, z^k)\| \leq c_k L_{\psi_1} \|(x^{k+1}, y^{k+1}, z^{k+1}) - (x^k, y^k, z^k)\|.$$

Using the inequality (33) and Lemma A.5, we have

$$
\begin{aligned}
\|\nabla_{xy}\psi_{c_k}(x^k, y^k, z^k) - c_k(d_x^k, d_y^k)\| &= c_k \left\|\nabla_{xy}\phi_{c_k}(x^k, y^k, z^k) - (d_x^k, d_y^k)\right\| \\
&\leq c_k C_{\psi_1} \left\|(\theta^{k+1}, \lambda^{k+1}) - (\theta_r^*(x^k, y^k, z^k), \lambda_r^*(x^k, y^k, z^k))\right\| \\
&\leq c_k C_{\psi_1} \left\|(\theta^k, \lambda^k) - (\theta_r^*(x^k, y^k, z^k), \lambda_r^*(x^k, y^k, z^k))\right\|,
\end{aligned}
$$
(57)

with $C_{\psi_1} := \sqrt{\max\{2(L_f + C_Z L_{g_1})^2 + 1/\gamma_1^2, 2L_g^2\}}$. Hence, we have

$$
\begin{aligned}
\|e_{xy}^k\| &\leq c_k L_{\psi_1} \|(x^{k+1}, y^{k+1}, z^{k+1}) - (x^k, y^k, z^k)\| + \frac{c_k}{\alpha_k} \left\|(x^{k+1}, y^{k+1}) - (x^k, y^k)\right\| \\
&\quad + c_k C_{\psi_1} \left\|(\theta^k, \lambda^k) - (\theta_r^*(x^k, y^k, z^k), \lambda_r^*(x^k, y^k, z^k))\right\|.
\end{aligned}
$$

For $\|e_z^k\|$, we have

$$\|e_z^k\| \leq \|\nabla_z\psi_{c_k}(x^{k+1}, y^{k+1}, z^{k+1}) - c_k d_z^k\| + \frac{c_k}{\beta_k} \left\|z^{k+1} - z^k\right\|.$$

Using Lemmas A.1 and A.5, we have

$$
\begin{aligned}
\|\nabla_z\psi_{c_k}(x^{k+1}, y^{k+1}, z^{k+1}) - c_k d_z^k\| &= c_k \left\|-\nabla_z v_{\gamma,r}(x^{k+1}, y^{k+1}, z^{k+1}) - d_z^k\right\| \\
&\leq \frac{c_k}{\gamma_2} \left(\left\|\lambda^{k+1} - \lambda_r^*(x^{k+1}, y^{k+1}, z^{k+1})\right\| + \left\|z^{k+1} - z^k\right\|\right) \\
&\leq \frac{c_k}{\gamma_2} \left(\left\|\lambda^k - \lambda_r^*(x^{k+1}, y^{k+1}, z^{k+1})\right\| + \left\|z^{k+1} - z^k\right\|\right).
\end{aligned}
$$

Therefore, we have

$$\|e_z^k\| \leq \frac{c_k}{\beta_k} \left\|z^{k+1} - z^k\right\| + \frac{c_k}{\gamma_2} \left(\left\|\lambda^k - \lambda_r^*(x^{k+1}, y^{k+1}, z^{k+1})\right\| + \left\|z^{k+1} - z^k\right\|\right).$$

With the estimations of $\|e_{xy}^k\|$ and $\|e_z^k\|$, we obtain the existence of $L_\psi > 0$ such that

$$
\begin{aligned}
R_k(x^{k+1}, y^{k+1}, z^{k+1}) &\leq c_k L_\psi \|(x^{k+1}, y^{k+1}, z^{k+1}) - (x^k, y^k, z^k)\| \\
&\quad + \frac{c_k}{\alpha_k} \left\|(x^{k+1}, y^{k+1}) - (x^k, y^k)\right\| + \frac{c_k}{\beta_k} \left\|z^{k+1} - z^k\right\| \\
&\quad + c_k \left(C_{\psi_1} + \frac{1}{\gamma_2}\right) \left\|(\theta^k, \lambda^k) - (\theta_r^*(x^k, y^k, z^k), \lambda_r^*(x^k, y^k, z^k))\right\|.
\end{aligned}
$$

Utilizing this inequality, let $\alpha_k \geq \underline{\alpha}$ and $\beta_k \geq \underline{\beta}$ for some positive constants $\underline{\alpha}, \underline{\beta}$, we can show that there exists $C_R > 0$ such that

$$
\begin{aligned}
&\frac{1}{c_k^2} R_k(x^{k+1}, y^{k+1}, z^{k+1})^2 \\
&\leq C_R \left(\frac{1}{4\alpha_k} \|(x^{k+1}, y^{k+1}) - (x^k, y^k)\|^2 + \frac{1}{4\beta_k}\|z^{k+1} - z^k\|^2 \right. \\
&\qquad\qquad \left. + \underline{\eta}\rho_T C_{\theta\lambda} \left\|(\theta^k, \lambda^k) - (\theta_r^*(x^k, y^k, z^k), \lambda_r^*(x^k, y^k, z^k))\right\|^2\right).
\end{aligned}
$$
(58)

Combining this with the inequality (52) implies that

$$\sum_{k=0}^{\infty} \frac{1}{c_k^2} R_k(x^{k+1}, y^{k+1}, z^{k+1})^2 < \infty.$$
(59)

Because $2p < 1$, it holds that

$$\sum_{k=0}^{K} \frac{1}{c_k^2} = \frac{1}{\underline{c}^2} \sum_{k=0}^{K} \left(\frac{1}{k+1}\right)^{2p} \geq \frac{1}{\underline{c}^2} \int_1^{K+2} \frac{1}{t^{2p}} dt \geq \frac{(K+2)^{1-2p} - 1}{(1-2p)\underline{c}^2},$$

and we can conclude from the inequality (59) that

$$\min_{0 \le k \le K} R_k(x^{k+1}, y^{k+1}, z^{k+1}) = O\left(\frac{1}{K^{(1-2p)/2}}\right).$$

Thus we complete the proof of the estimate (50).

Finally, since $\psi_{c_k}(x^k, y^k, z^k) \le M$ and $F(x^k, y^k) \ge \underline{F}$ for any $k$, we have

$$c_k\left(f(x^k, y^k) - v_{\gamma,r}(x^k, y^k, z^k)\right) \le M - \underline{F}, \quad \forall k,$$

and we can obtain from $c_k = \underline{c}(k+1)^p$ that

$$f(x^k, y^k) - v_{\gamma,r}(x^k, y^k, z^k) = O\left(\frac{1}{K^p}\right).$$

Since $v_{\gamma,r}(x, y, z) \le v_\gamma(x, y, z) \le f(x, y)$ for all $(x, y) \in C$, we get

$$0 \le f(x^k, y^k) - v_\gamma(x^k, y^k, z^k) \le f(x^k, y^k) - v_{\gamma,r}(x^k, y^k, z^k) = O\left(\frac{1}{K^p}\right).$$

This establishes the desired estimate (51), completing the proof. □

