# OpenReview forum: "Constrained Bi-Level Optimization: Proximal Lagrangian Value Function Approach and Hessian-free Algorithm"
_ICLR.cc/2024/Conference — ICLR 2024 spotlight_

### Official Review · Reviewer_4Vih · 2023-10-22

**Soundness:** 2 fair
**Presentation:** 3 good
**Contribution:** 3 good
**Rating:** 5
**Confidence:** 4

**Summary:**

This paper concerns bi-level optimization (BLO) problems with coupled inner constraints. By introducing the Moreau envelope of the Lagrange function, the lower-level problem can be reformulated into a smooth function value constraint. Then, a single loop algorithm is proposed based on the formulation. Non-asymptotic rate is established for the proposed method using a newly defined stationarity measure.

**Strengths:**

1. The smoothing technique based on the Moreau envelope is good. This provides a new reformation for the BLO.

2. The theoretical rate and numerical experiments are sufficient.

**Weaknesses:**

1.Assumption 3.1(i) seems too strong for me. Usually, the involved functions in BLO are assumed convex only w.r.t. $y$.

2. The stationarity measure needs further clarification. The proposed measure is based on (16). However, it is not clear how the stationary points of (16) are related to the original BLO.

3. The techniques in the theoretical proofs lack novelty, which directly takes advantage of analysis for penalty methods.

**Questions:**

What is the central benefit of using the Moreau envelope instead of the optimal value function, especially in the technical parts?

**Details Of Ethics Concerns:**

This submission is rather similar to submission 3145 of ICLR 2024. The similarities between submission 3552 (this paper) and 3145 are summarized as follows.

(a)Submissions 3552 and 3145 both consider BLO with constrained LL problems.

(b)Submission 3552 (denoted by [A]) considers the proximal functions in dealing with the LL problem and submission 3145 (denoted by [B]) uses the Moreau envelope. These two techniques are basically the same in terms of the inner sights. Please compare (7) (8) in [A] and (7) (8) in [B].

(c)The main equations in the two submissions are highly similar. Please compare (9)-(16) in [A] and (9)-(16) in [B].

(d)The theoretical results are basically the same. Please compare Theorem 3.1 in [A] and Theorem 3.1 in [B].

I hope that the authors can address my concern. Thanks.

---

> ### Author Response · Authors · 2023-11-17
> **Response to Reviewer 4Vih**
>
> We sincerely appreciate the time you dedicated to reviewing our paper. Your feedback is valuable to us, and we would like to address some specific points for clarification.
>
> (1) Assumption 3.1(i) seems too strong for me. Usually, the involved functions in BLO are assumed convex only w.r.t. $y$.
>
> $\textbf{Reply:}$
> We appreciate your comment and would like to clarify. First, Assumption 3.1 is about the $L$-smoothness and the lower boundedness of $F(x,y)$, without imposing any convexity condition on $F$.
>
> If you are referring to the convexity in Assumption 3.3(i), it is a common and standard assumption in constrained bilevel optimization. Recent research, such as Gao et al. (2023), also employs this assumption.
>
> It is important to note that in all experiments conducted by Khanduri et al. (2023), Xiao et al. (2023b), and Gao et al. (2023), as well as in the application models used in our numerical experiments, the lower-level constraints are either linear or linearly coupled and thus consistently satisfy Assumption 3.3(i).
>
> (2) The stationarity measure needs further clarification. The proposed measure is based on (16). However, it is not clear how the stationary points of (16) are related to the original BLO.
>
> $\textbf{Reply:}$
> To provide some clarifications:
> Firstly, the equivalence between the constrained problem (6) and the original BLO problem (1) is rigorously discussed in Theorems A.1 and A.2. Additionally, in our study, we utilize the residual function $R_k$, as defined in (15), as the measure of stationarity for the constrained problem (6). This approach is justified as $R_k$ acts as a stationarity measure for the approximate KKT condition of problem (6), as discussed in Andreani et al. (2010). Moreover, $R_k$ is also applicable as a stationarity measure for the penalized problem (16) of problem (6).
>
> (3) The techniques in the theoretical proofs lack novelty, which directly takes advantage of analysis for penalty methods.
>
> $\textbf{Reply:}$
> Our work introduces significant technical innovations for several reasons:
>
> (i)  We diverge from traditional methods by proposing a novel single-level equivalent reformulation for constrained BLO problems. This reformulation is based on a newly introduced proximal Lagrangian value function associated with the constrained LL problem.
>
> (ii)  The newly introduced proximal Lagrangian value function is defined as the value function of a strongly-convex-strongly-concave proximal min-max problem. Our study of its continuous differentiability, along with the derivation of its derivative as presented in Lemma A.1, constitutes a novel theoretical contribution.
>
> (iii)  Additionally, our findings regarding the Lipschitz continuity of the saddle point solution to the strongly-convex-strongly-concave proximal min-max problem defining the proximal Lagrangian value function, as detailed in Lemma A.3, represent a significant theoretical advancement.
>
> (iv)  Furthermore, the formulation of the descent lemma for the proximal Lagrangian value function, as elaborated in Lemmas A.2 and A.4, also signifies a theoretical progression.
>
> (v)  The newly established properties of the proximal Lagrangian value function, as mentioned above, play a crucial role in underpinning the non-asymptotic convergence of our proposed single-loop algorithm.
>
> (4) What is the central benefit of using the Moreau envelope instead of the optimal value function, especially in the technical parts?
>
> $\textbf{Reply:}$
> Firstly, we would like to clarify that our approach is not the Moreau envelope of the lower-level problem as described in Gao et al. (2023). Our new proposed algorithm is based on the concept of the proximal Lagrangian value function.
>
> In constrained BLO, methods based on value functions face challenges due to the non-differentiability of constraints arising from such value function-based reformulations. This issue arises even when the lower-level problem's data is smooth, as the value function typically lacks differentiability if the solutions or the Lagrangian multipliers of the lower-level problem are not unique.
>
> To address this challenge, we introduce a new proximal Lagrangian value function tailored for handling constrained LL problem. Utilizing this function, we propose a new single-level reformulation for constrained BLOs, transforming them into equivalent single-level optimization problems with smooth constraints. The proximal Lagrangian value function, defined as the value function of a strongly-convex-strongly-concave min-max problem, exhibits the advantageous property of continuous differentiability.
>
> Building on this reformulation, our work develops a Hessian-free gradient-based algorithm for constrained BLOs and provides a non-asymptotic convergence analysis.

---

> > ### Author Response · Authors · 2023-11-23
> >
> > (2)$\textbf{Differences in Main Ideas and the Design of Algorithms:}$
> >
> > To address the differnet challenges encountered in BLO problems with unconstrained nonconvex nonsmooth lower-level problem and constrained convex lower-level problem, Submission 3145 and Submission 3552 developed different approaches and proposed distinct algorithms with using the same essential idea as in the value function-based reformulation, which is recently popularly applied and study in recent bilevel optimization works, see, e.g., Liu et al. (2021b; 2023a); Ye et al. (2022); Sow et al. (2022); Shen & Chen (2023); Kwon et al. (2023a); Lu & Mei (2023).
> >
> > Submission 3145, utilized $\textbf{Moreau envelope-based}$ reformulation of BLO, originally presented in Gao et al. (2023), combined with the idea of single-loop bilevel optimization in previous works, to design a single-loop and Hessian-free gradient-based algorithm. Thanks to the Moreau envelope-based reformulation and the strong convexity proximal term in the Moreau envelope, even when the convexity of lower-level problem or the PL condition usually required by other single-loop bilevel optimization works is absent, the error induced by the one step iteration of lower-level problem can controled. And adding a proximal operator of the nonsmooth component step in the iteration scheme also makes the propsed algorithm be able to handle the nonsmooth of the lower-level problem.
> >
> >
> >
> > In Submission 3552, inspired by the idea of proposing new value function to replace the value function in the value function approach reformulation in Gao et al. (2023),  a novel $\textbf{proximal Lagrangian value function}$ is introduced to handle constrained LL problem. By leveraging this function, a new single-level reformulation for constrained BLOs, converting them into equivalent single-level optimization problems with smooth constraints is proposed. The reason that Submission 3552 proposes a new proximal Lagrangian value function instead of uses Moreau envelope-based reformulation for handling constrained LL problem with coupled lower-level constraints is that as shown in Theorem 2.3 in Gao et al. (2023), the Moreau envelope of the constrained LL problem, as well as the value function of the constrained LL problem lacks differentiability if the solutions or the Lagrangian multipliers of the lower-level problem are not unique. However, the differentiability of the “value” function is very important for designing single-loop bilevel optimization algorithm, especially for controlling the error induced by the one step iteration of lower-level problem.
> >
> > To overcome this differentiability issue, with focusing on the convex LL problem scenario, Submission 3552 propose using the proximal Lagrangian value function, defined as the value function of a strongly-convex-strongly-concave min-max problem, exhibits the advantageous property of continuous differentiability, see Lemma A.1 within the Appendix of Submission3552. Since the problems and methods being addressed are different, the design of the algorithms also varies.

---

> > ### Author Response · Authors · 2023-11-23
> >
> > (3)$\textbf{Differences in Theoretical Analysis:}$
> >
> > The proof frameworks for non-asymptotic convergence results in submissions 3145 (Theorem 3.1) and 3552 (Theorem 3.1) are similar to those of prior single-loop bilevel optimization works, as evidenced by their foundational approach to establishing a sufficient descent inequality for the merit function and subsequently deriving non-asymptotic estimations from such inequalities.
> >
> > To the best of our knowledge, this theoretical analysis approach for BLO originated from Ghadimi & Wang (2018), and has been widely employed in establishing the convergence rates of various algorithms, including stocBiO (Ji et al., 2021), ALSET (Chen et al., 2021), TTSA (Hong et al., 2023), AmIGO ( Arbel & Mairal, 2022a), FSLA (Li et al., 2022), SOBA and SABA (Dagréou et al., 2022), $F^2SA$ (Kwon et al., 2023a), V-PBGD (Shen & Chen, 2023) and others.
> >
> > Notwithstanding the structural similarity of the merit functions in submissions 3145 and 3552 to earlier works, these submissions introduce new elements due to the distinct problems, reformulations, and algorithms they encompass. Specifically, the hyper-objective components and the lower-level problem error terms, resulting from a one-step iteration strategy in prior works, are replaced by unique penalized objectives and corresponding iteration error terms in these submissions.
> >
> > (i) However, the different problem settings and reformulations in submissions 3145 and 3552 engender substantial differences in the techniques, arguments for the proof of a critical element—the sufficient descent inequality of the merit function—for their non-asymptotic convergence results, as elaborated in submissions 3145 (Lemma 3.1) and 3552 (Lemma 3.1). These disparities are primarily manifest in the proofs of auxiliary results.
> >
> > (ii) The methodologies and arguments employed in these auxiliary proofs differ markedly between the two submissions. In submission 3145, the proof of the descent lemma for $-v_\gamma(x,y)$ (Lemma A.3) hinges on the weak convexity of $v_\gamma(x,y)$, leveraging the properties of the Moreau envelope to obviate the need for convexity in the lower-level problem. Conversely, submission 3552 confronts a more complex scenario with its proximal Lagrangian value function $v_{\gamma,r}(x,y,z)$, defined as a min-max problem. The direct establishment of descent lemmas for $-v_{\gamma,r}(x,y,z)$ is notably challenging, prompting the derivation of variant descent lemmas for $-v_{\gamma,r}(x,y,\bar{z})$ and $-v_{\gamma,r}(\bar{x}, \bar{y}, z)$, with fixed $(\bar{x}, \bar{y})$ and $\bar{z}$ (Lemma A.2 and A.4). These variants require intricate techniques tailored to the min-max problem and also depend on the convexity of the lower-level problem.
> >
> > (iii) The demonstration of Lipschitz continuity for $\theta^*_\gamma(x,y)$ in Lemma A.5 of submission 3145 and for $(\theta^*(x,y,z), \lambda^*(x,y,z))$ in Lemma A.3 of submission 3552 also exhibit pronounced differences. In submission 3145, addressing the non-smooth term in the lower-level problem necessitates employing a fixed-point characterization of $\theta^*_\gamma(x,y)$. This is combined with leveraging the Lipschitz property with respect to $x$ of the proximal operator for the non-smooth component $g(x,y)$ to establish the Lipschitz continuity of $\theta^*_\gamma(x,y)$. Conversely, in submission 3552, the pair $(\theta^*(x,y,z), \lambda^*(x,y,z))$ represents a saddle point in a min-max problem, necessitating a recharacterization of this saddle point as a solution to a generalized equation. Subsequently, the application of monotone operator theorem results is crucial to proving their Lipschitz continuity.
> >
> > (iv) Regarding Lemma A.6 in submission 3145 and Lemma A.3 in submission 3552, the contrast in the generation of $\theta^{k+1}$ through a proximal gradient step in a single-level optimization problem, and $(\theta^{k+1}, \lambda^{k+1})$ through a forward-backward step in a min-max problem, necessitates distinct approaches in their contraction property proofs.
> >
> > $\textbf{In summary}$, while submissions 3145 and 3552 adhere to the overarching proof structures of non-asymptotic convergence results characteristic of prior single-loop bilevel optimization studies, they necessitate significantly varied and specialized techniques. These adaptations address specific challenges and complexities arising from non-convexity and non-smoothness in the lower-level problem of the bilevel programs considered in submission 3145, and the lower-level coupled constraints in the bilevel programs considered  in submission 3552, which are pivotal in proving non-asymptotic convergence.

---

> > ### Author Response · Authors · 2023-11-23
> >
> > (4)$\textbf{Differences in Experiments:}$
> >
> > Submission3145 validate the effectiveness and efficiency of the proposed algorithm MEHA on various bilevel optimization problems with nonconvex lower-level problem, such as, few-shot learning, data hyper-cleaning and the real-world neural architecture search application.
> >
> >
> > Submission3552 evaluate the efficiency of the proposed algorithm LV-HBA through numerical experiments on various bilevel optimization problems with lower-level problem with couples constraints, such as synthetic problems, hyperparameter optimization for SVM and federated bilevel learning.

---

> ### Author Response · Authors · 2023-11-23
> **Response to Reviewer 4Vih**
>
> We respectfully disagree with Reviewer 4Vih's comments regarding the Ethics Concerns, which did not appear in the original version of this reviewer’s Official Review. It should be noted that, as indicated by the system time record, these newly added Ethics Concerns were included in the reviewer’s Official Review only after we had submitted our rebuttal to their initial review. Moreover, the reviewer did not respond to our rebuttal. Consequently, we did not receive any notifications from the system and only became aware of these newly added Ethics Concerns a few hours ago.
>
> Below, we provide detailed clarifications to address this biased comments.
>
>
>
> (1)   $\textbf{Differences in Problem Settings and Challenges:}$
>
> It's important to note that submissions 3145 and 3552 are tackling two distinct challenges that arise in the development of single-loop Hessian-free algorithms for bi-level optimization (BLO) problems. Specifically, the configurations of the BLO problems, and more specifically, the settings of the lower-level problems examined in submissions 3145 and 3552, are significantly different.
>
> To be more specific, submission 3145 focuses on BLO with an $\textbf{unconstrained}$ lower-level problem with potentially $\textbf{nonconvex}$ objective and a potentially additional $\textbf{nonsmooth}$ component depending on both upper- and lower-level variables and it is for addressing the nonconvexity and nonsmothness challenges appeared in the lower-level problem of BLO.
>
> In contrast, submission3552 addressed BLO problems with  $\textbf{constrained}$ convex lower-level problems.To provide more detail, the lower-level problem is with coupled lower-level constraints, and both the objective and constraint functions of lower-level problem is smooth and convex with respect to the lower-level variable. And submission3552 primarily addressed challenges arising from coupled lower-level constraints, i.e., lower-level $x$-dependent constraints.
>
>
> To summarize, the differences in problem settings and challenges can be outlined as follows.
> |                  | Submission3552 | Submission3145  |
> | :--------: | :--------: | :-------------: |
> |   $\textbf{Lower-level problem}$ | $\min_{y\in Y} f(x,y) \ \mathrm{s.t.}\  g(x,y)\leq 0$      |       $\min_{y\in Y} \varphi(x,y):=f(x,y)+g(x,y) $       |
> |   $\textbf{Annotation}$ | Here $g(x,y)$ is a coupled lower-level constraint      |       Here $g(x,y) $ is the nonsmooth part of the lower-level objective       |
> |   $\textbf{Challenges}$ | Lower-level $x$-dependent constraints, and non-differentiability in both value function and Moreau envelope-based reformulations      |       nonconvexity and nonsmoothness in LL objective      |

---

> ### Comment · Area_Chair_VUZz · 2023-12-05
>
> Dear Reviewer 4Vih
>
> Thank you so much for pointing out the potential research integrity issue (dual submission). I realize Reviewer W8M5 also share his/her opinion on this, so I response there with my thought. Please take a look at that discussion and leave your thoughts.
>
> Thanks,
>
> AC

---

### Official Review · Reviewer_W8M5 · 2023-10-28

**Soundness:** 3 good
**Presentation:** 3 good
**Contribution:** 4 excellent
**Rating:** 8
**Confidence:** 5

**Summary:**

This paper studied a class of new constrained bilevel optimization, where the lower-level problem involves constraints coupling both upper-level and lower-level variables. Based on the Moreau envelope value function, this paper proposed an efficient single-loop Hessian-free gradient-based algorithm. Moreover, it studied the non-asymptotic convergence analysis for the proposed algorithm. Extensive experimental results verify the efficiency of the proposed algorithm. In summary, the contributions of this paper are significant on the design algorithm and solid theoretical analysis.

**Strengths:**

This paper studied a class of new constrained bilevel optimization, where the lower-level problem involves constraints coupling both upper-level and lower-level variables. Based on the Moreau envelope value function, this paper proposed an efficient single-loop Hessian-free gradient-based algorithm. Moreover, it studied the non-asymptotic convergence analysis for the proposed algorithm. Extensive experimental results verify the efficiency of the proposed algorithm. In summary, the contributions of this paper are significant on the design algorithm and solid theoretical analysis.

**Weaknesses:**

It is better to list some bilevel optimization examples in machine learning that have non-smooth and weakly convex lower level functions, which will strength the motivation of this work.

**Questions:**

Some questions:

1)	In the proposed LV-HBA algorithm, there exist many hyper parameters such as $\alpha_k,\beta_k,\eta_k,\gamma,c_k$. Although the authors provided the range of these parameters in the convergence analysis, I still think that the choice of these hyper parameters is not easy in practice.

2)	In the experiments, how to choose these hyper parameters including the parameter $r$ in set $Z$ ?

3)	From the Theorem 3.1, the best convergence rata is $O(1/K^{1/4})$ when $p=1/4$ ?

4)	It is better to list some bilevel optimization examples in machine learning that have nonsmooth and weakly convex lower level functions, which will strength the motivation of this work.

---

> ### Author Response · Authors · 2023-11-17
> **Response to Reviewer W8M5. Part 1**
>
> We sincerely appreciate the time you dedicated to reviewing our paper and are grateful for the valuable insights you provided. In the following, we will address each of your questions with thorough consideration.
>
> (1) In the proposed LV-HBA algorithm, there exist many hyper parameters such as $\alpha_k$, $\beta_k$, $\eta_k$, $\gamma$, $c_k$. Although the authors provided the range of these parameters in the convergence analysis, I still think that the choice of these hyper parameters is not easy in practice.
>
> $\textbf{Reply:}$
> In addition to the step size parameters, our method involves three hyperparameters: $\gamma$, $r$, and $c_k$. The parameters $\gamma$ and $r$ define the truncated proximal Lagrangian value function. Theoretically, the requirements for $\gamma$ and $r$ are modest; $\gamma$ is positive, and $r$ should be large. The penalty parameter, $c_k$, follows a specific selection strategy outlined in Theorem 3.1, namely $c_k = \underline{c}(k+1)^p$ where $p \in (0,1/2)$ and $\underline{c}>0$. Consequently, in practice, typically only the step size parameters require careful tuning.
>
> In our numerical experiments, we analyzed the sensitivity of the parameter $c_k$, with its convergence curve displayed in Figure 2.
>
> Additionally, to assess the sensitivity of the remaining parameters, further numerical experiments were conducted on the LL merely convex synthetic model. We record the time when $\|x^k - x^*\| / \|x^*\| \leq 10^{-2}$ is met by the iterates generated by LV-HBA. The results are as follows:
>
> |    Strategy     | &nbsp;&emsp;$α$ |   $β$   |   $η$   | $γ=γ_1=γ_2$ | $\underline{c}$ |  $p$   | $r$    | Time(s) |
> | :-------------: | :-----------: | :---: | :---: | :-----: | :-------------: | :--: | ---- | ------- |
> |        $α$        |     0.001     | 0.02  | 0.01  |   10    |      0.025      | 0.3  | 1000 | 0.23    |
> |        $α$        |     0.005     | 0.02  | 0.01  |   10    |      0.025      | 0.3  | 1000 | 0.064   |
> |        $α$        |     0.001     | 0.02  | 0.01  |   10    |      0.025      | 0.3  | 1000 | 0.052   |
> |        $β$        |     0.005     | 0.005 | 0.01  |   10    |      0.025      | 0.3  | 1000 | 0.068   |
> |        $β$        |     0.005     | 0.02  | 0.01  |   10    |      0.025      | 0.3  | 1000 | 0.064   |
> |        $β$        |     0.005     |  0.1  | 0.01  |   10    |      0.025      | 0.3  | 1000 | 0.063   |
> |        $η$        |     0.005     | 0.02  | 0.005 |   10    |      0.025      | 0.3  | 1000 | 0.098   |
> |        $η$        |     0.005     | 0.02  | 0.01  |   10    |      0.025      | 0.3  | 1000 | 0.064   |
> |        $η$        |     0.005     | 0.02  | 0.05  |   10    |      0.025      | 0.3  | 1000 | 0.025   |
> |        $γ=γ_1=γ_2$        |     0.005     | 0.02  | 0.01  |    5    |      0.025      | 0.3  | 1000 | 0.069   |
> |        $γ=γ_1=γ_2$        |     0.005     | 0.02  | 0.01  |   10    |      0.025      | 0.3  | 1000 | 0.064   |
> |        $γ=γ_1=γ_2$        |     0.005     | 0.02  | 0.01  |   500   |      0.005      | 0.3  | 1000 | 0.064   |
> | $\underline{c}$ |     0.005     | 0.02  | 0.01  |   10    |      0.005      | 0.3  | 1000 | 0.026   |
> | $\underline{c}$ |     0.005     | 0.02  | 0.01  |   10    |      0.05       | 0.3  | 1000 | 0.064   |
> | $\underline{c}$ |     0.005     | 0.02  | 0.01  |   10    |      0.025      | 0.3  | 1000 | 0.086   |
> |        $r$        |     0.005     | 0.02  | 0.01  |   10    |      0.025      | 0.3  | 200  | 0.066   |
> |        $r$        |     0.005     | 0.02  | 0.01  |   10    |      0.025      | 0.3  | 1000 | 0.064   |
> |        $r$        |     0.005     | 0.02  | 0.01  |   10    |      0.025      | 0.3  | 2000 | 0.065   |

---

> ### Author Response · Authors · 2023-11-17
> **Response to Reviewer W8M5. Part 2**
>
> (2) In the experiments, how to choose these hyper parameters including the parameter $r$ in set $Z$?
>
> $\textbf{Reply:}$
> Thank you for your question. The table below outlines the parameter settings for the experiments involving the parameter $r$:
> |      Expriment       | &nbsp;&emsp; $α$ |   $β$   |  $η$   |  $γ=γ_1=γ_2$   |    $r$   | $\underline{c}$ |  $p$   |
> | :------------------: | :-----------: | :---: | :--: | :--: | :--: | :-------------: | :--: |
> |         LL merely convex          |     0.002     | 0.002 | 0.03 |  10  |  1   |        1        | 0.3  |
> |         LL strongly convex         |     0.01      | 0.01  | 0.05 |  10  | 1000 |        1        | 0.3  |
> |         SVM          |     0.01      |  0.1  | 0.01 |  10  | 200  |       10        | 0.3  |
> | Data Hyper-Cleaning |     0.01      |  0.1  | 0.01 |  10  | 200  |       10        | 0.3  |
>
>
>
> (3) From the Theorem 3.1, the best convergence rata is $O(1/K^{1/4})$ when $p=1/4$?
>
> $\textbf{Reply:}$
> Yes, you are correct.
>
> (4) It is better to list some bilevel optimization examples in machine learning that have nonsmooth and weakly convex lower level functions, which will strength the motivation of this work.
>
> $\textbf{Reply:}$
> Thank you for your suggestions. Since the $L$-smoothness of $f$ over $X \times Y$ ensures its weak convexity of $f$ on the same domain, the setting of a weakly convex lower-level objective encompasses all sufficiently smooth applications within bounded $X \times Y$ in the realm of machine learning.

---

> > ### Comment · Reviewer_W8M5 · 2023-11-23
> > **Reply to Authors**
> >
> > Thanks for your responses. My concerns have been well-addressed, so I keep my score.

---

### Official Review · Reviewer_hK2o · 2023-10-30

**Soundness:** 3 good
**Presentation:** 3 good
**Contribution:** 3 good
**Rating:** 8
**Confidence:** 3

**Summary:**

This paper introduces a novel gradient-based algorithm designed for a class of constrained Bi-Level Optimization (BLO) problems that have coupled lower-level (LL) constraints. In this problem, the lower-level problem takes the form:
$$
\min_y f(x,y), s.t. g(x,y) \leq 0.
$$
In contrast to existing methods relying on implicit differentiation techniques, this paper introduces a smooth proximal Lagrangian value function for the lower-level problem. This new value function enables the formulation of a single-level reformulation for the constrained BLO. Building upon this reformulation, a single-loop alternating gradient descent method is derived to efficiently tackle the constrained BLO. The paper establishes a non-asymptotic convergence result for the proposed algorithm. Additionally, the paper includes numerical experiments conducted on both illustrative toy examples and practical applications, demonstrating the algorithm's superior practical performance.

**Strengths:**

1.	This paper is well-organized and in general easy to follow. The assumptions concerning the functions and problem settings are clearly elucidated, facilitating a clear grasp of both the primary concept and the technical intricacies of the proposed approach.

2.	A noteworthy contribution to the bi-level optimization community is the introduction of the proximal Lagrangian value function and the resulting single-level reformulation for constrained BLOs. This addition holds significant promise for advancing the development of other efficient methods to the constraints BLOs.

3.	Notably, the proposed method distinguishes itself by its Hessian-free nature, obviating the need for any computations involving the Hessian matrix of the lower-level problem data. Furthermore, its single-loop structure renders it highly implementable and computationally efficient. An appealing aspect of this method is its ability to work without imposing the stringent requirement of strong convexity on the lower-level problem. Moreover, it exhibits versatility by accommodating scenarios in which the lower-level problem possesses multiple solutions, expanding its applicability to a broad spectrum of practical applications.

5.	The paper extensively explores the properties of the newly introduced proximal Lagrangian value function and the associated reformulation. Additionally, a non-asymptotic convergence analysis is included, further substantiating the contribution and the validity of the proposed approach.

**Weaknesses:**

1.	In Table 1, to the best of my knowledge, BVFSM does not require LL objective to be convex.

2.	The authors have not included some recent works that investigate algorithms for addressing constrained Bi-Level Optimization (BLO) problems through the value function approach, such as:

Fliege, J., Tin, A., & Zemkoho, A. (2021). Gauss–Newton-type methods for bilevel optimization. Computational Optimization and Applications, 78(3), 793-824.

Fischer, A., Zemkoho, A. B., & Zhou, S. (2022). Semismooth Newton-type method for bilevel optimization: global convergence and extensive numerical experiments. Optimization Methods and Software, 37(5), 1770-1804.

3.	On page 8,  $\|y^k-y^*(x)\|$ should be revised to `` $\left\|y^k-y^*(x^k)\right\| $ "?

4.	When the constraints of the LL problem are absent, i.e., when $g = 0$, does the introduced proximal Lagrangian value function revert to the classical Moreau envelope function of the LL objective? Furthermore, is there any relationship between the proximal Lagrangian value function and the augmented Lagrangian function of the LL problem?

**Questions:**

I think this work provides some good contributions to constrained bilevel optimization, and hence I tend to accept it. See my questions in the weakness part.

---

> ### Author Response · Authors · 2023-11-17
> **Response to Reviewer hK2o**
>
> We sincerely appreciate the time and effort you dedicated to reviewing our paper and are grateful for your constructive comments. Below, we will respond to each of the questions you raised.
>
> (1) In Table 1, to the best of my knowledge, BVFSM does not require LL objective to be convex.
>
> $\textbf{Reply:}$
> Thank you for your observation. You are correct that BVFSM does not require the LL objective to be convex. We appreciate your comment and will clarify this point in a future revision of the paper.
>
>
> (2) The authors have not included some recent works that investigate algorithms for addressing constrained Bi-Level Optimization (BLO) problems through the value function approach, such as:
>
> Fliege, J., Tin, A., & Zemkoho, A. (2021). Gauss–Newton-type methods for bilevel optimization. Computational Optimization and Applications, 78(3), 793-824.
>
> Fischer, A., Zemkoho, A. B., & Zhou, S. (2022). Semismooth Newton-type method for bilevel optimization: global convergence and extensive numerical experiments. Optimization Methods and Software, 37(5), 1770-1804.
>
> $\textbf{Reply:}$
> Thank you for sharing these recent works with us. We'll take them into account when revising our manuscript.
>
>
> (3) On page 8, $|y^k - y^*(x)|$ should be revised to $|y^k - y^*(x^k)|$?
>
> $\textbf{Reply:}$
> Thank you for pointing out the typo. We'll correct it in the revised version.
>
> (4) When the constraints of the LL problem are absent, i.e., when $g=0$, does the introduced proximal Lagrangian value function revert to the classical Moreau envelope function of the LL objective? Furthermore, is there any relationship between the proximal Lagrangian value function and the augmented Lagrangian function of the LL problem?
>
> $\textbf{Reply:}$
> Thank you for your interesting questions.
>
> Firstly, your observation is accurate: the proximal Lagrangian value function introduced in our study simplifies to the classical Moreau envelope function of the lower-level objective in the absence of LL problem constraints. Regarding the connection between this proximal Lagrangian value function and the augmented Lagrangian function of the LL problem, this indeed presents an intriguing avenue for exploration. We intend to investigate this relationship in our future research endeavors.

---

### Official Review · Reviewer_ttDx · 2023-10-31

**Soundness:** 3 good
**Presentation:** 3 good
**Contribution:** 3 good
**Rating:** 6
**Confidence:** 2

**Summary:**

This paper introduces a new single-loop Hessian-free algorithm for the solving Bi-Level Optimization (BLO) problems.

It first creates a smooth proximal Lagrangian value function, effectively addressing the constrained lower-level problem. Subsequently, the authors present a single-level reformulation for constrained BLOs, converting the original BLO problem into an equivalent optimization problem with smooth constraints.

The paper includes a non-asymptotic convergence analysis for the proposed algorithm. Some experiments have been conducted to show the superior practical performance of the algorithm

**Strengths:**

S1. This paper proposes the first single-loop Hessian-free algorithm for solving the BLO problem.
S2. The authors introduce a new potential function, which is associated with the monotonically decreasing step size. Furthermore, they demonstrate how to select these step sizes to ensure the potential function exhibits the sufficient descent property.
S3. The authors have conducted experiments on five machine learning tasks to validate the performance of their proposed methods.

**Weaknesses:**

W1. There are an excessive number of stepsize parameters in use, and it remains uncertain whether the algorithm's performance is significantly affected by the choice of these parameters.

W2. While other bilevel optimization algorithms employ techniques such as Nesterov's momentum or utilize high-order information of the objective function, the plain and simple gradient descent/ascent algorithm may be slow in practical applications.

W3. Assumption 3.3 (ii) can be quite stringent since the global Lipschitz constant for both $L_{g_1}$ and $L_{g_2}$ have not been determined in advance for many applications.

W4. The choice for the parameter $r$ for the truncated proximal Lagrangian value function is not mentioned.

**Questions:**

See above.

---

> ### Author Response · Authors · 2023-11-17
> **Response to Reviewer ttDx. Part 1**
>
> We sincerely appreciate the time and effort you devoted to reviewing our paper. Your valuable feedback is greatly appreciated, and we would like to clarify some specific points in response.
>
> (1) There are an excessive number of stepsize parameters in use, and it remains uncertain whether the algorithm's performance is significantly affected by the choice of these parameters.
>
> $\textbf{Reply:}$
> In addition to the step size parameters, our method involves three hyperparameters: $\gamma$, $r$, and $c_k$. The parameters $\gamma$ and $r$ define the truncated proximal Lagrangian value function. Theoretically, the requirements for $\gamma$ and $r$ are modest; $\gamma$ is positive, and $r$ should be large. The penalty parameter, $c_k$, follows a specific selection strategy outlined in Theorem 3.1, namely $c_k = \underline{c}(k+1)^p$ where $p \in (0,1/2)$ and $\underline{c}>0$. Consequently, in practice, typically only the step size parameters require careful tuning.
>
> In our numerical experiments, we analyzed the sensitivity of the parameter $c_k$, with its convergence curve displayed in Figure 2.
>
> Additionally, to assess the sensitivity of the remaining parameters, further numerical experiments were conducted on the LL merely convex synthetic model. We record the time when $\|x^k - x^*\| / \|x^*\| \leq 10^{-2}$ is met by the iterates generated by LV-HBA. The results are as follows:
>
> |    Strategy     | &nbsp;&emsp;$α$ |   $β$   |   $η$   | $γ=γ_1=γ_2$ | $\underline{c}$ |  $p$   | $r$    | Time(s) |
> | :-------------: | :-----------: | :---: | :---: | :-----: | :-------------: | :--: | ---- | ------- |
> |        $α$        |     0.001     | 0.02  | 0.01  |   10    |      0.025      | 0.3  | 1000 | 0.23    |
> |        $α$        |     0.005     | 0.02  | 0.01  |   10    |      0.025      | 0.3  | 1000 | 0.064   |
> |        $α$        |     0.001     | 0.02  | 0.01  |   10    |      0.025      | 0.3  | 1000 | 0.052   |
> |        $β$        |     0.005     | 0.005 | 0.01  |   10    |      0.025      | 0.3  | 1000 | 0.068   |
> |        $β$        |     0.005     | 0.02  | 0.01  |   10    |      0.025      | 0.3  | 1000 | 0.064   |
> |        $β$        |     0.005     |  0.1  | 0.01  |   10    |      0.025      | 0.3  | 1000 | 0.063   |
> |        $η$        |     0.005     | 0.02  | 0.005 |   10    |      0.025      | 0.3  | 1000 | 0.098   |
> |        $η$        |     0.005     | 0.02  | 0.01  |   10    |      0.025      | 0.3  | 1000 | 0.064   |
> |        $η$        |     0.005     | 0.02  | 0.05  |   10    |      0.025      | 0.3  | 1000 | 0.025   |
> |        $γ=γ_1=γ_2$        |     0.005     | 0.02  | 0.01  |    5    |      0.025      | 0.3  | 1000 | 0.069   |
> |        $γ=γ_1=γ_2$        |     0.005     | 0.02  | 0.01  |   10    |      0.025      | 0.3  | 1000 | 0.064   |
> |        $γ=γ_1=γ_2$        |     0.005     | 0.02  | 0.01  |   500   |      0.005      | 0.3  | 1000 | 0.064   |
> | $\underline{c}$ |     0.005     | 0.02  | 0.01  |   10    |      0.005      | 0.3  | 1000 | 0.026   |
> | $\underline{c}$ |     0.005     | 0.02  | 0.01  |   10    |      0.05       | 0.3  | 1000 | 0.064   |
> | $\underline{c}$ |     0.005     | 0.02  | 0.01  |   10    |      0.025      | 0.3  | 1000 | 0.086   |
> |        $r$        |     0.005     | 0.02  | 0.01  |   10    |      0.025      | 0.3  | 200  | 0.066   |
> |        $r$        |     0.005     | 0.02  | 0.01  |   10    |      0.025      | 0.3  | 1000 | 0.064   |
> |        $r$        |     0.005     | 0.02  | 0.01  |   10    |      0.025      | 0.3  | 2000 | 0.065   |

---

> ### Author Response · Authors · 2023-11-17
> **Response to Reviewer ttDx. Part 2**
>
> (2) While other bilevel optimization algorithms employ techniques such as Nesterov's momentum or utilize high-order information of the objective function, the plain and simple gradient descent/ascent algorithm may be slow in practical applications.
>
> $\textbf{Reply:}$
> In the numerical experiments section, we compare our method against two competitors: AiPOD and E-AiPOD (Xiao et al., 2023b), as well as GAM (Xu & Zhu, 2023). Both of these are bilevel optimization algorithms that leverage the Hessian information from the lower-level objective. Our results demonstrate a significant advantage of our method: it relies solely on gradient information of the lower-level problem. This approach not only simplifies the computational process but also notably enhances the speed, rendering our method considerably faster than the competing algorithms.
>
> (3) Assumption 3.3 (ii) can be quite stringent since the global Lipschitz constant for both $L_{g_1}$ and $L_{g_2}$ have not been determined in advance for many applications.
>
> $\textbf{Reply:}$
> We appreciate your feedback and understand your concern about Assumption 3.3 (ii). Here are some clarifications:
>
> (i) The Lipschitz continuity of the gradient of the lower-level constraints in Assumption 3.3 (ii) is a common and standard assumption in the context of constrained bi-level optimization. This assumption is similarly employed in recent studies, such as those by Khanduri et al. (2023) and Xiao et al. (2023b).
>
> (ii) Importantly, in practical applications considered in the experiments conducted in Khanduri et al. (2023) and Xiao et al. (2023b), including scenarios like Adversarially Robust Learning and Federated Bilevel Learning, the lower-level constraints are either linear or linearly coupled. This observation holds true for all models assessed in our numerical experiments. In such cases, the global Lipschitz constants for both $L_{g_1}$ and $L_{g_2}$ are equal to zero.
>
> (4) The choice for the parameter $r$ for the truncated proximal Lagrangian value function is not mentioned.
>
> $\textbf{Reply:}$
> As demonstrated in Theorem A.2, that given a solution $(x^*, y^*, z^*)$ to the reformulation (4), selecting the parameter $r$ to be sufficiently large, specifically $r \ge \|z^*\|_{\infty}$, ensures that $(x^*, y^*, z^*)$ also solves variant (6), Therefore, it is standard to choose a large value for the parameter $r$ in practice.

---

### Official Review · Reviewer_7k2H · 2023-11-11

**Soundness:** 2 fair
**Presentation:** 3 good
**Contribution:** 2 fair
**Rating:** 5
**Confidence:** 3

**Summary:**

This paper presents a single loop proximal Lagrangian Value function-based Hessian-free Bi-level Algorithm (LV-HBA) for solving constrained bilevel optimization where the lower-level problem involves constraints coupling both upper-level and lower-level variables.
The authors relax the strongly convex assumption on lower-level problem to the general convex case, and provide non-asymptotic convergence analysis for LV-HBA. They also provide numerical experiments on the synthetic problem, hyperparameter selection problem and data hypercleaning task.

**Strengths:**

The paper is well-written. The authors propose a single loop Hessian free method that we refer to as Lagrangian Value function-based Hessian-free Bi-level Algorithm (LV-HBA). The authors provide non-asymptotic convergence analysis for LV-HBA, relaxing the underlying assumptions for lower level problem from strongly convexity to only convexity.

**Weaknesses:**

Please see questions.

**Questions:**

1. Can the authors elaborate why there is no need to assume the Lipschitz continuity of the upper level function $F(x,y)$, which is typically necessary in bilevel optimization even the lower function is strongly convex, e.g. Xiao et al. (2023b).
2. On page 6, the sentences after Assumption 3.2 say $f$ is $\rho_f$-weakly convex on $X\times Y$ with $\rho_f\ge 0$, which is potentially being smaller than $L_f$. Can you provide the exact form of $\rho_f$ or show the case of $\rho_f$ being smaller for some specific application? Does that mean we can only suppose $\rho_f = L_f$ in general?
3. Theorem A.1 says reformulation (4) is equivalent to constrained BLO problem (1). In the proof of Theorem A.1, it only illustrates that the feasible points of (4) and (1) are equivalent. It is unclear why this means the formulations (4) and (1) and are also equivalent. Can we conclude the optimal solution of (4) and (1) are equivalent?
4. In reformulation (5), how to choose the parameter $r$?
5. The proposed methods includes many hyperparameters, making difficult to implement in practice.

---

> ### Author Response · Authors · 2023-11-17
> **Response to Reviewer 7k2H. Part 1**
>
> We appreciate your time and effort in reviewing our work and providing constructive feedback. Below, we will address each of your questions with careful consideration.
>
> (1) Can the authors elaborate why there is no need to assume the Lipschitz continuity of the upper level function $F(x,y)$, which is typically necessary in bilevel optimization even the lower function is strongly convex, e.g. Xiao et al. (2023b).
>
> $\textbf{Reply:}$
> Our approach diverges fundamentally from previous algorithms, such as the one proposed by Xiao et al. (2023b). Specifically, Xiao et al. assume the strong convexity of the lower-level objective with respect to $y$, leading to a unique solution for the lower-level problem, denoted as $y^*(x)$. Their algorithm is designed based on the hyper-gradient of the hyper-objective $F(x, y^*(x))$. The Lipschitz continuity of the upper-level objective $F(x, y)$ is utilized to establish the $L$-smoothness of the hyper-objective.
>
> In contrast, our work does not presume strong convexity of the lower-level objective with respect to $y$. This allows for the possibility of multiple lower-level solutions, and consequently, the hyper-objective framework used by Xiao et al. is inapplicable. Instead of relying on the hyper-objective, our paper introduces an innovative single-level reformulation for constrained bilevel optimization. This reformulation translates the problem into an equivalent single-level optimization problem with a smooth constraint. We then develop a gradient-type method to address this reformulation. Consequently, our theoretical analysis requires only that $F(x,y)$ be $L_F$-smooth, as detailed in Assumption 3.1, without necessitating the Lipschitz continuity of $F(x, y)$.
>
> (2) On page 6, the sentences after Assumption 3.2 say $f$ is $\rho_f$-weakly convex on $X\times Y$ with $\rho_f\geq0$, which is potentially being smaller than $L_f$. Can you provide the exact form of  or show the case of $\rho_f$ being smaller for some specific application? Does that mean we can only suppose $\rho_f=L_f$ in general?
>
> $\textbf{Reply:}$
> A typical instance occurs when $f$ is convex over the domain $X \times Y$. In such scenarios, $\rho_f$ equals $0$, which is less than $L_f$. This property appears in bilevel optimization problems related to hyperparameter selection, where $f$ often exhibits convexity. For detailed examples, refer to the following works:
>
> Lucy L Gao, Jane J. Ye, Haian Yin, Shangzhi Zeng, and Jin Zhang. Value function based difference-of-convex algorithm for bilevel hyperparameter selection problems. In ICML, 2022.
>
> Jane J. Ye, Xiaoming Yuan, Shangzhi Zeng, and Jin Zhang. Difference of convex algorithms for bilevel programs with applications in hyperparameter selection. Mathematical Programming, 198(2):1583–1616, 2023.
>
>
> (3) Theorem A.1 says reformulation (4) is equivalent to constrained BLO problem (1). In the proof of Theorem A.1, it only illustrates that the feasible points of (4) and (1) are equivalent. It is unclear why this means the formulations (4) and (1) and are also equivalent. Can we conclude the optimal solution of (4) and (1) are equivalent?
>
> $\textbf{Reply:}$
> In addition to the equivalence of feasible points between formulation (4) and (1), they share identical objectives, denoted as $F(x,y)$. Consequently, both (4) and (1) have identical optimal solutions, applicable in both global and local contexts.
>
> (4) In reformulation (5), how to choose the parameter $r$?
>
> $\textbf{Reply:}$
> As demonstrated in Theorem A.2, that given a solution $(x^*, y^*, z^*)$ to the reformulation (4), selecting the parameter $r$ to be sufficiently large, specifically $r \ge \|z^*\|_{\infty}$, ensures that $(x^*, y^*, z^*)$ also solves variant (6). Therefore, it is standard to choose a large value for the parameter $r$ in practice.

---

> ### Author Response · Authors · 2023-11-17
> **Response to Reviewer 7k2H. Part 2**
>
> (5) The proposed methods includes many hyperparameters, making difficult to implement in practice.
>
> $\textbf{Reply:}$
> We appreciate your feedback and would like to clarify a few points:
>
> In addition to the step size parameters, our method involves three hyperparameters: $\gamma$, $r$, and $c_k$. The parameters $\gamma$ and $r$ define the truncated proximal Lagrangian value function. Theoretically, the requirements for $\gamma$ and $r$ are modest; $\gamma$ is positive, and $r$ should be large. The penalty parameter, $c_k$, follows a specific selection strategy outlined in Theorem 3.1, namely $c_k = \underline{c}(k+1)^p$ where $p \in (0,1/2)$ and $\underline{c}>0$. Consequently, in practice, typically only the step size parameters require careful tuning.
>
> In our numerical experiments, we analyzed the sensitivity of the parameter $c_k$, with its convergence curve displayed in Figure 2.
>
> Additionally, to assess the sensitivity of the remaining parameters, further numerical experiments were conducted on the LL merely convex synthetic model. We record the time when $\|x^k - x^*\| / \|x^*\| \leq 10^{-2}$ is met by the iterates generated by LV-HBA. The results are as follows:
>
> |    Strategy     | &nbsp;&emsp;$α$ |   $β$   |   $η$   | $γ=γ_1=γ_2$ | $\underline{c}$ |  $p$   | $r$    | Time(s) |
> | :-------------: | :-----------: | :---: | :---: | :-----: | :-------------: | :--: | ---- | ------- |
> |        $α$        |     0.001     | 0.02  | 0.01  |   10    |      0.025      | 0.3  | 1000 | 0.23    |
> |        $α$        |     0.005     | 0.02  | 0.01  |   10    |      0.025      | 0.3  | 1000 | 0.064   |
> |        $α$        |     0.001     | 0.02  | 0.01  |   10    |      0.025      | 0.3  | 1000 | 0.052   |
> |        $β$        |     0.005     | 0.005 | 0.01  |   10    |      0.025      | 0.3  | 1000 | 0.068   |
> |        $β$        |     0.005     | 0.02  | 0.01  |   10    |      0.025      | 0.3  | 1000 | 0.064   |
> |        $β$        |     0.005     |  0.1  | 0.01  |   10    |      0.025      | 0.3  | 1000 | 0.063   |
> |        $η$        |     0.005     | 0.02  | 0.005 |   10    |      0.025      | 0.3  | 1000 | 0.098   |
> |        $η$        |     0.005     | 0.02  | 0.01  |   10    |      0.025      | 0.3  | 1000 | 0.064   |
> |        $η$        |     0.005     | 0.02  | 0.05  |   10    |      0.025      | 0.3  | 1000 | 0.025   |
> |        $γ=γ_1=γ_2$        |     0.005     | 0.02  | 0.01  |    5    |      0.025      | 0.3  | 1000 | 0.069   |
> |        $γ=γ_1=γ_2$        |     0.005     | 0.02  | 0.01  |   10    |      0.025      | 0.3  | 1000 | 0.064   |
> |        $γ=γ_1=γ_2$        |     0.005     | 0.02  | 0.01  |   500   |      0.005      | 0.3  | 1000 | 0.064   |
> | $\underline{c}$ |     0.005     | 0.02  | 0.01  |   10    |      0.005      | 0.3  | 1000 | 0.026   |
> | $\underline{c}$ |     0.005     | 0.02  | 0.01  |   10    |      0.05       | 0.3  | 1000 | 0.064   |
> | $\underline{c}$ |     0.005     | 0.02  | 0.01  |   10    |      0.025      | 0.3  | 1000 | 0.086   |
> |        $r$        |     0.005     | 0.02  | 0.01  |   10    |      0.025      | 0.3  | 200  | 0.066   |
> |        $r$        |     0.005     | 0.02  | 0.01  |   10    |      0.025      | 0.3  | 1000 | 0.064   |
> |        $r$        |     0.005     | 0.02  | 0.01  |   10    |      0.025      | 0.3  | 2000 | 0.065   |

---

### Meta-Review · Area_Chair_VUZz · 2023-12-06

**Metareview:**

This research studies a new type of constrained bi-level optimization problem, where the lower-level problem includes constraints. The study introduces an effective single-loop algorithm that doesn't rely on Hessian calculations and is based on the Moreau envelope value function.

The paper makes notable contributions in terms of developing the algorithm and providing a thorough theoretical analysis. The paper provides extensive experimental data to demonstrate the effectiveness of the algorithm.

The stationarity measure needs further clarification. In particular, how are the stationary points of (16) are related to the original BLO.

**Justification For Why Not Higher Score:**

The stationarity measure needs further clarification. In particular, how are the stationary points of (16) are related to the original BLO. It seems that the authors approximate the original problem and then only focus on finding a stationary point on the approximate problem. It will be good to study the relationship between the stationary points  of the original and the approximate problem.

**Justification For Why Not Lower Score:**

The paper makes notable contributions in terms of developing the algorithm and providing a thorough theoretical analysis. The paper provides extensive experimental data to demonstrate the effectiveness of the algorithm.

---

### Decision · Program_Chairs · 2024-01-16

Accept (spotlight)